# A GIS-Based Spatiotemporal Impact Assessment of Droughts in the Hyper-Saline Urmia Lake Basin on the Hydro-Geochemical Quality of Nearby Aquifers

**Bakhtiar Feizizadeh [1,2,*], Zahra Abdollahi [3] and Behzad Shokati [4]**

[1] Applied GISciences Lab, Department of Geography, Humboldt-Universität zu Berlin, 10781 Berlin, Germany
[2] Department of Remote Sensing and GIS, University of Tabriz, Tabriz 51368, Iran
[3] Soil Conservation and Watershed Management Research Department, Zanjan Agricultural and Natural Resources Research and Education Center, AREEO, Zanjan 45131, Iran; abdollahi_zhr@yahoo.com
[4] Young Researchers and Elite Club, Maragheh Branch, Islamic Azad University, Maragheh 55131, Iran; shokati.behzad@yahoo.com
[*] Correspondence: feizizadeh@tabrizu.ac.ir; Tel.: +98-09143058630

**Abstract:** Urmia Lake is a hyper-saline lake in northwestern Iran that has been drying up since 2005. The main objective of this study was to evaluate the water quality in aquifers that are the main source of fresh water for the eastern plains Urmia Lake, which has been drying up due to intensive land use/cover changes and climate change. We evaluated hydro-geochemical data and factors contributing to aquifer pollution and quality variation for nine aquifers in the vicinity of Urmia Lake during the dry and wet seasons from 2000–2020. Our methodology was based on the analysis of 10 years of data from 356 deep and semi-deep wells using GIS spatial analysis, multivariate statistical analysis, and agglomerative hierarchical clustering. We developed a Water Quality Index (WQI) for spatiotemporal assessment of the status of the aquifers. In doing so, we highlighted the value of combining Principal Component Analysis (PCA), WQI, and GIS to determine the hydro-geochemical attributes of the aquifers. We found that the groundwater in central parts of the study area was unsuitable for potable supplies. Anthropogenic sources of contamination, such as chemical fertilizers, industrial waste, and untreated sewage water, might be the key factors causing excessive concentrations of contaminants affecting the water quality. The PCA results showed that over 80% of the total variance could be attributed to two principal factors for most aquifers and three principal factors for two of the aquifers. We employed GIS-based spatial analysis to map groundwater quality in the study area. Based on the WQI values, approximately 48% of groundwater samples were identified as poor to unsuitable for drinking purposes. Results of this study provide a better hydro-geochemical understanding of the multiple aquifers that require preventive action against groundwater damage. We conclude that the combined approach of using a multivariate statistical technique and spatial analysis is effective for determining the factors controlling groundwater quality.

**Keywords:** aquifer degradation; groundwater extraction; hydrochemistry; Urmia Lake; Principal Component Analysis; Water Quality Index

## 1. Introduction

Groundwater, which makes up approximately one-third of the freshwater on earth, is an indispensable water source for irrigation, industrial purposes, and domestic needs [1]. Groundwater also plays a significant role in maintaining the water levels of rivers, lakes, and wetlands [2]. It is the main source of water for agricultural and industrial activities as well as drinking needs for human settlement areas in arid and semi-arid environments around the world. Iran, a mostly arid country with limited water resources, has suffered from extended droughts for decades [3]. Groundwater abstraction ranges between 20 and 53 km$^3$ per year and accounts for 55% of Iran's water use, of which 92% is used

for agricultural purposes [4]. Increasing demands, global climate change, and a lack of continual improvement in resource management are imposing unsustainable pressure on the country's aquifers [5].

Overexploitation of groundwater leads to diminishing water quantities, which leads to water quality degradation [6,7]. As groundwater is extracted for various purposes, contaminants become increasingly concentrated in the remaining water and deteriorate the water quality. Contamination is simply the presence of a substance where it should not be or at concentrations above background, but is not yet proven that it can result in adverse biological effects on resident communities. It is well understood that excessive pumping and discharge of aquifers can cause saltwater and contaminants to migrate upward [8]. Such an issues leads to tangible environmental issues such as groundwater salinization, which threatens human livelihood by means of impacting the drinking water, food production, etc. Groundwater quality is determined based on aquifer minerals, regional geology, climatic conditions, and anthropogenic activities [9]. Michel (2017) reported that Iran's water resources received 70.74 points out of 100 on the global Water Quality Index (WQI) scale. Although it is above the global mean of 67 [10], several basins face significant pollution due to domestic wastewater and agricultural runoff. Iran has six key catchment areas [5]. Groundwater levels are reported to be declining by 3.1–11.2 mm per year in each of the six major basins [11].

As a unique socio-ecological region in northwestern Iran, Urmia Lake Basin (ULB) also bears increasing strain because of persistent management deficiencies and climate change [8,12]. A decreasing groundwater table, reduced water levels in streams and lakes, and deteriorating water quality are some effects of groundwater depletion in this basin [13]. Early studies monitoring land use and cover changes reported that agricultural activities and farmland areas have doubled during the last 30 years [14,15]. Furthermore, the number of wells has increased, and consequently, so has groundwater exploitation around the lake [16]. Over the past few decades, several regional hydrogeological studies have analyzed hydro-geochemical data from aquifers to better understand the changes in groundwater quality [17]. Previously, several hydrology and hydro-geochemistry studies have been carried out in different parts of the ULB [8,13,18–20]. However, no systematic investigation and spatial-temporal analysis of hydro-geochemical data have been carried out along the eastern shore of Urmia Lake.

The hydro-geochemical datasets, routinely collected from wells, are often large, imprecise, non-normally distributed, and complex, making them difficult to visualize. The Principal Component Analysis (PCA) has recently been applied to simplify geochemical groundwater chemistry datasets with remarkable success [21–24]. The PCA interprets the variance of large datasets of correlated variables and summarizes the information content into a smaller set of uncorrelated variables known as their principal component [24]. Early studies indicated that the application of statistical analysis can help us to better understand the characteristics of different aquifers with complex data matrices [25–28]. Agglomerative Hierarchical Cluster Analysis (ACA) analysis can also be used to categorize observations into two or more groups based on similarities between the samples according to a set of particular characteristics to make the results easier to understand and interpret [29].

The Water Quality Index (WQI) is an indicator widely used to express the water quality of a water body by transforming the list of variables and their concentrations into a single value. The term WQI was first introduced by Horton as an efficient index for analyzing the quality of aquifers [30]. The WQI method has been widely applied to assess the quality of both surface water and groundwater [2,28–32]. From an environmental perspective, groundwater salinization may have significant environmental issues. The accumulation of water-soluble salts in the upper soil layers causes salinization, which significantly impacts agricultural productivity, environmental health, and economic wellbeing [33]. As previously mentioned, groundwater is often a major source of drinking water, particularly in arid and semi-arid environments. It has been considered a major source of drinking water in urban and rural areas located in the ULB. Due to intensive land use and cover changes,

and drought as a result of climate change, the hyper-salty Urmia Lake has been drying up, which has resulted in intensive environmental issues such as soil salinization, salt/dust storm scattering, and food shortage [14,34]. Based on this statement, the main objective of this study was to evaluate the water quality of nine aquifers as the main resource of drinking water in the eastern plains of Urmia Lake using a GIS-based multivariate statistical analysis. In order to observe the impact of lake drought on the salinization of nearby aquifers and drinking water resources, we also traced water quality variation between aquifers and seasons to delineate the factors responsible for aquifer salinization and declining water quality based on their geographical location in the ULB.

## 2. Study Area

Urmia Lake is a hyper-saline lake in northwestern Iran (Figure 1). The lake has been drying up since 2005 because of the extensive land-use changes, expanding farmlands, human settlements, and industrial activities [13,19]. Urmia Lake Basin (ULB) is a center of agricultural activities, human settlements, and intensive industrial activities. Cities and settlements east of Urmia Lake, such as Tabriz, Azarshahr, Bonab, Maleken, Maraqeh, Ajabshir, Marand, and Shabestar, are surrounded by very productive agricultural farmlands that produce food for the 7.9 million inhabitants living in the ULB. Groundwater has been one of the main drinking water resources for the local population and agricultural activities

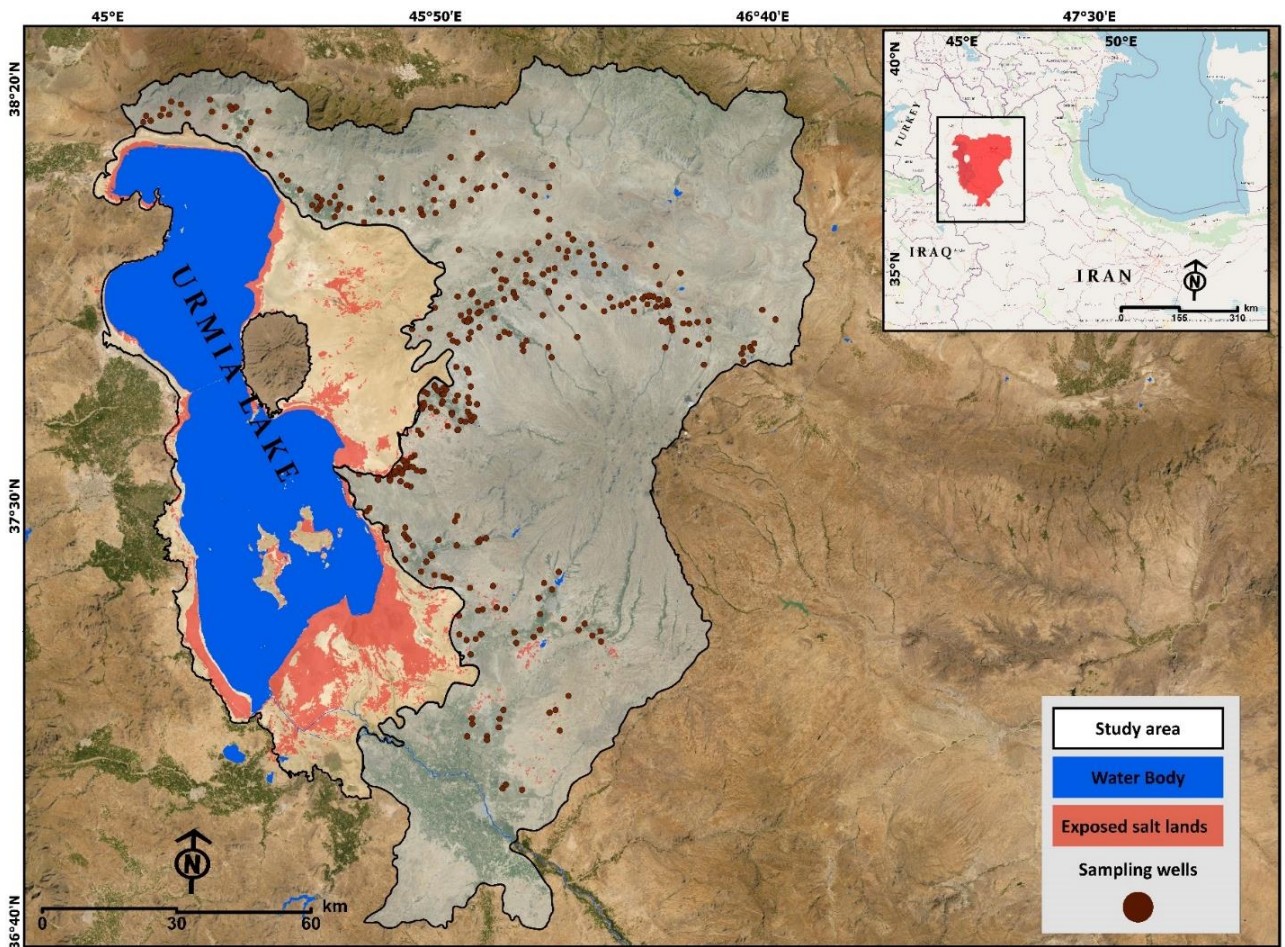

**Figure 1.** Location of the study area and the sampling wells.

The ULB is characterized by a cold semi-arid climate and a Mediterranean Continental climate. The mean annual temperature is 12 °C, which drops to between −10 and −15 °C in the winter. The annual precipitation is approximately 320 mm [35]. Figure 2 shows the annual prescription and temperature trends for the ULB. Groundwater supplies most of the

water for drinking and domestic use in urban and rural areas. Accordingly, due to minimal precipitation from June to September, all domestic water consumption, and agricultural and industrial activities rely on groundwater, which creates intensive pressure on the aquifers. The traditional agricultural system comprises small family-owned farms that cultivate crops with predominantly high water demand (e.g., onions, melons, watermelons, alfalfa, apples, grapes, etc.) [34,36] using irrigation dependent on groundwater (Figure 3). Climate change has led to drought and intensive agricultural activities has placed extensive pressure on regional groundwater resources [37,38]. Land subsidence has occured as a result of intensive groundwater abstraction and aquifer salinization. The resulting interaction between the hyper-saline water of the lake and the freshwater in nearby aquifers can be observed around the lake [38]. The groundwater extraction was reported to be about 1.2 billion cubic meters in 2012, which indicates an over-extraction of groundwater in the study region (Iran Water Resources Management). According to water balance results, the Shabestar-Sufian and Tasuj aquifers have the highest reservoir deficits [39].

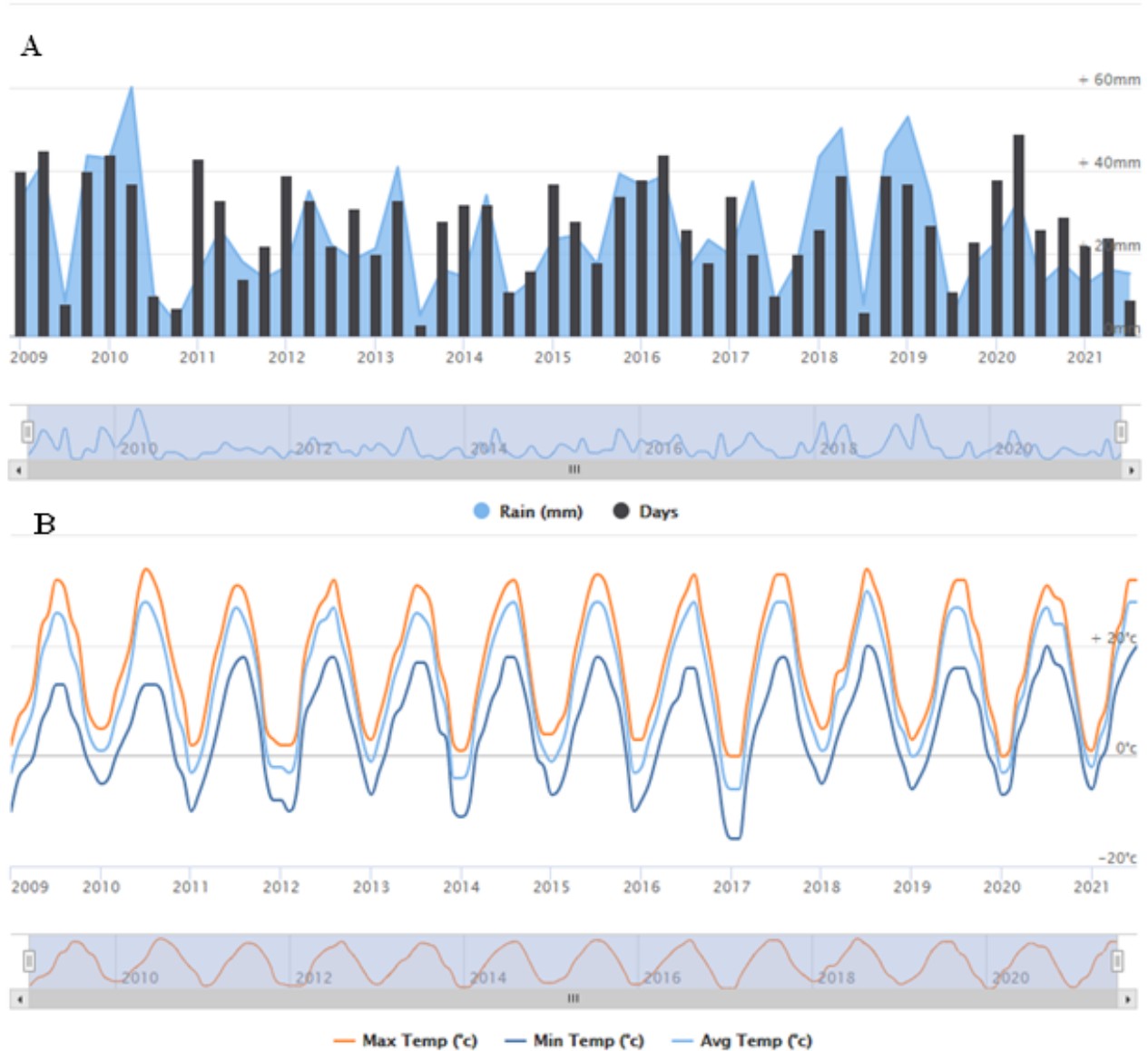

**Figure 2.** Annual precipitation (**A**) and temperature (**B**) trend for the ULB From 2009–2021. Annual precipitation in the ULB has been decreasing while the minimum temperatures have increased.

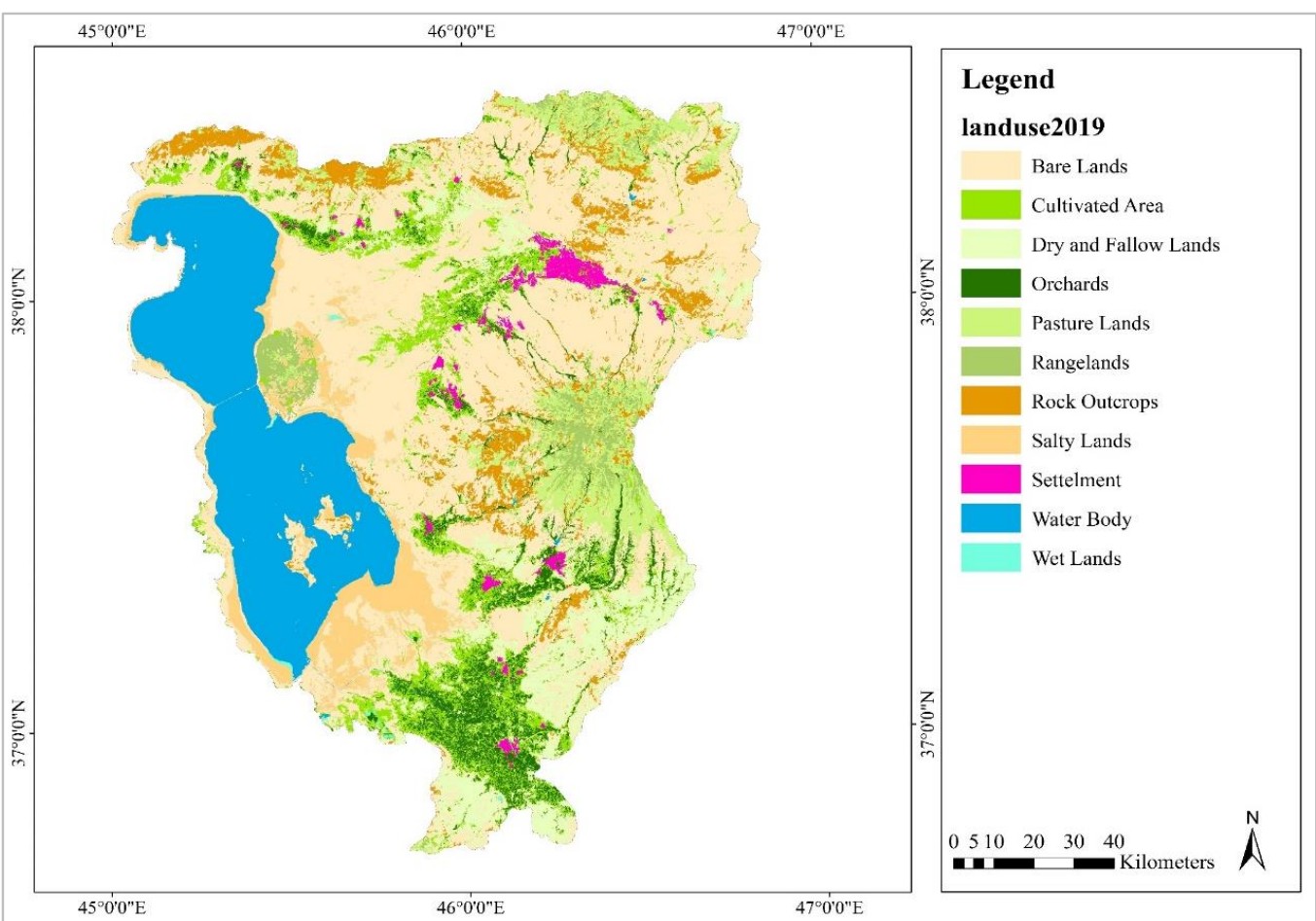

**Figure 3.** Land-use map of the study area for 2020.

## 3. Materials and Methods

### 3.1. Geological Setting of the ULB

The rather complex geology of the ULB is the result of several tectonic events throughout geological history. The many thermal springs all over the basin attest to the extent of volcanic activity. The metamorphic rocks and Paleozoic platform sediments (a part of the Central Iran zone) north of the lake are bound by the Tabriz-Sofian-Bostan Abad fault to the north and east [40]. The volcanic sedimentary rocks, such as travertine and young traces, which occur extensively in the southeast of the Azarshahr plain, compose the aquifers that yield the primary source of groundwater for agriculture [8]. The Sabalan Volcano (4810 m), consisting of volcanic ash rocks, and the salt domes in Aji Chai are the major geological features in northeastern Azerbaijan, northeast of the Tabriz fault (in the Alborz-Azerbaijan zone). Salt clay is also observed around the lake [8]. Geologically, we can distinguish between unpressurized and pressurized aquifers in the basin. The unpressurized aquifer within the Plio-Pleistocene tuffs on the northern slope of the Sahand is highly developed in terms of water resources [41]. A complete geological map is shown in Figure 4.

### 3.2. Dataset

We received quarterly water quality monitoring results for the ULB from the East Azerbaijan Province Regional Water Organization. This includes hydro-chemical monitoring data from 356 observation wells, comprising shallow to deep wells, springs, and qanats (traditional wells). The well data has been collected since 2000, which enabled us to identify groundwater quality issues affecting the eastern plains of Urmia Lake. Therefore, we analyzed groundwater quality parameters, viz. power of hydrogen (pH), electrical conduc-

tivity (EC), total dissolved solids (TDS), total hardness (TH), sodium ($Na^+$), potassium ($K^+$), calcium ($Ca^{2+}$), magnesium ($Mg^{2+}$), chloride ($Cl^-$), sulfate ($SO_4^{2-}$), carbonate ($CO_3^{2-}$), and bicarbonate ($HCO_3^-$) for the study period of 2000–2020. We also used elevation data derived from topographic maps and geological maps of the ULB at a scale of 1/25,000. The land use/cover map of the study area was derived from Landsat satellite images using the integrated approach of fuzzy object based image analysis and deep learning methods, which was published in one of our previous studies [37,42].

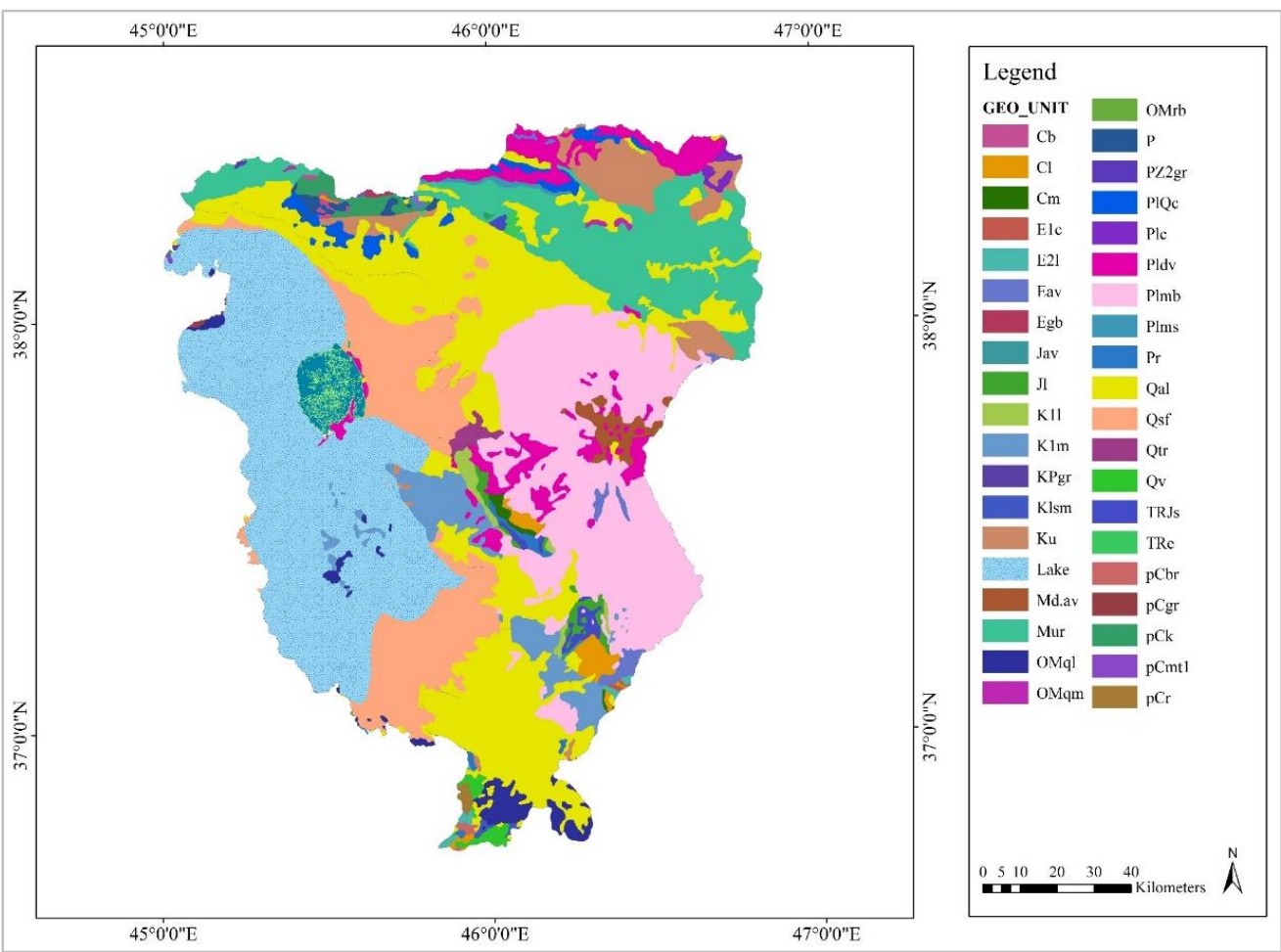

**Figure 4.** Geological map of the study area.

### 3.3. Water Quality Index Implementation

The Water Quality Index (WQI) is an indicator that reduces large datasets to a single value to reflect the condition of water and its suitability for drinking [31]. To calculate the WQI of each well during the study period, we selected 11 water quality parameters (pH, EC, TDS, TH, $HCO_3^-$, $SO_4^{2-}$, $Cl^-$, $Ca^{2+}$, $Mg^{2+}$, $K^+$, and $Na^+$). Each parameter was assigned a weighting (wi) between 1 and 5 according to its relative importance to overall water quality (Table 1). Different weights were attributed to each parameter, depending on the use, study region, climate, and water body. Thus, with the definition of weights, the water quality parameters can be used to calculate the index as comprehensively as possible [43].

**Table 1.** Weight and relative weight of each parameter used to calculate the WQI and water quality classification ranges and types of water based on Water Quality Index (WQI) values.

| Parameter | Unit | WHO's Standard (Si) | Weight of Each Parameter (wi) | Relative Weight (Wi) |
|---|---|---|---|---|
| $SO_4{}^{2-}$ | mg/L | 250 | 4 | 0.121 |
| $Cl^-$ | mg/L | 250 | 3 | 0.091 |
| $HCO_3{}^-$ | mg/L | 500 | 3 | 0.091 |
| pH | - | 6.5–8.5 | 4 | 0.121 |
| EC | us/cm | 500 | 4 | 0.121 |
| TDS | mg/L | 500 | 5 | 0.152 |
| TH | mg/L | 500 | 3 | 0.091 |
| $K^+$ | mg/L | 12 | 2 | 0.061 |
| $Na^+$ | mg/L | 200 | 2 | 0.061 |
| $Mg^{2+}$ | mg/L | 50 | 1 | 0.030 |
| $Ca^{2+}$ | mg/L | 75 | 2 | 0.061 |

| WQI Value | Quality Based on the WQI |
|---|---|
| Less than 50 | Excellent |
| 50–100 | Good |
| 100–200 | Poor |
| 200–300 | Very poor |
| More than 300 | Unsuitable for drinking purposes |

The *WQI* was computed for each well based on the following Equations (1)–(4) [16]:

$$W_i = \frac{w_i}{\sum_{i=1}^{n} w_i} \tag{1}$$

$$q_i = \frac{C_i}{S_i} \times 100 \tag{2}$$

$$SI_i = q_i \times W_i \tag{3}$$

$$WQI = \sum SI_i \tag{4}$$

where $W_i$ refers to the relative weight of each parameter and the sum of the weights for this index must be equal to 1; $w_i$ is the weight assigned to each parameter; $n$ is the number of chosen parameters; $q_i$ expresses the relative quality of each parameter; $C_i$ is the measured concentration of each parameter in the sample (mg/L); $S_i$ is the World Health Organization's (WHO's) standard for each parameter (mg/L); and $SI_i$ refers to the sub-index of the ith parameter.

*3.4. Data Analysis*

The scatter plots presented in Figure 5 show the range of values for each variable and give an indication of outliers, i.e., observations that appear to be inconsistent with the rest of the data. The mean, median, range, standard deviation, skewness, and kurtosis were calculated for the different quality parameters for each aquifer. We evaluated the obtained hydro-chemical groundwater data to determine the water quality using the Water Quality Index (WQI), Spearman correlation, Principal Component Analysis (PCA), and Agglomerative Hierarchical Clustering Analysis (ACA) in combination with GIS-based spatial analysis. The statistical evaluation of the hydro-chemical groundwater parameters helps us to determine the main factors controlling water quality variation over time [44]. We assessed the strength of the association between the variables and the direction of the relationship using the Spearman correlation test. The Spearman rank correlation, a

non-parametric test that does not carry any assumptions about the distribution of the data, was computed using Equation (5).

$$\rho = 1 - \frac{6 \sum d_i^2}{n(n^2 - 1)} \tag{5}$$

where $\rho$ is the Spearman rank correlation and defines the difference between the ranks of corresponding variables, and $n$ represents the number of observations. High coefficients represent a correlation between two parameters. A positive coefficient signifies similarity and harmony between the correlated parameters, and a negative coefficient indicates the variables are developing in opposite directions [45]. We also carried out an Agglomerative Hierarchical Cluster Analysis (ACA) of the groundwater data to identify specific patterns of similar observations within the studied variables. ACA is used to group observations into subsets based on hierarchy [46]. We used the agglomerative clustering method as it could be used to clearly visualize the data as distinct groups. We used the Euclidean distance to estimate the distance between observations. We also chose the Ward linkage method [46] to combine clusters because, unlike other methods of combination, it can be used to analyze the variance for clusters of quantitative variables [47]. The Principal Component Analysis (PCA) was then applied to clarify complex patterns in data matrices [46–50] to reveal relationships between physical and chemical parameters for the aquifer. Multivariate statistical techniques, such as the PCA, make it straightforward to visualize and summarize the data [50]. PCA is a widely used data reduction technique that extracts the important factors (in the form of components) from a large set of variables available in a dataset [51]. Principal components are linear combinations of original predictors presented as X1, X2, ... , Xn, which capture the maximum variance in the dataset. In this case, it can be written as Equation (6):

$$Zn = \Phi^1 nX^1 + \Phi^2 nX^2 + \Phi^3 nX^3 + \ldots + \Phi pnXp \tag{6}$$

where, Zn refers to nth principal component and $\Phi pn$ is the loading vector consisting of loadings ($\Phi^1 n$, $\Phi^2 n$, ... ) of the nth principal component. The sum of the square of the loadings is equal to 1. This is because a large number of loadings may lead to a large variance [51]. The first principal component ($Z^1$) results in a line in p dimensional space that is closest to the data. $Z^1$ specifies the direction of the highest variability in the data. Closeness is measured using the average squared Euclidean distance. $X^1$, $X^2$, ... , Xp are normalized predictors with a mean value of zero and a standard deviation of one. Zn is uncorrelated with $Zn^{+1}$ [52,53].

The importance of a component (Zn) for an observation (n) is defined by the square of the cosine of the angle from the right angle made with the origin, the observation, and its projection on the principal plane, which is computed as per Equation (7). Observations with a high squared cosine (i.e., close to 1) represent a component of higher importance [54].

$$cos_{i,l}^2 = \frac{a_{i,l}^2}{h_{i,g}^2} \tag{7}$$

where $h_{i,g}^2$ is the squared distance of an observation (*n*) to the origin and $a_{i,l}^2$ refers to the squared coordinates of the components. Components with a large value for $cos_{i,l}^2$ contribute a relatively large portion to the total distance and, therefore, these components are important for that observation. Squared cosines of variables reflect the representation quality of a variable on a component axis. If the correlation of the squared cosine with an axis is low, the position of the variable on this axis should not be interpreted to avoid interpretation errors due to projection effects. We considered squared cosines with absolute values of more than or equal to 0.4 for the explanation of the observed ions [46,55].

The PCA helps us find structures in chaotic and complex data to simplify the explanation of observations [56]. GIS-based spatial analysis was applied to quickly and efficiently

determine distribution characteristics and develop spatial maps of groundwater quality. While many interpolation methods are available to approximate other pixel values, we used the inverse distance weighted (IDW) spatial interpolation technique, which is the most commonly used deterministic model [57]. The theory behind this model is based on the hypothesis that a node has the greatest similarity to nearby points and is thus influenced more by nearby data values. Thus, spatial interpolation of the data from the 356 observation wells was calculated by averaging the weighted sum of all the points.

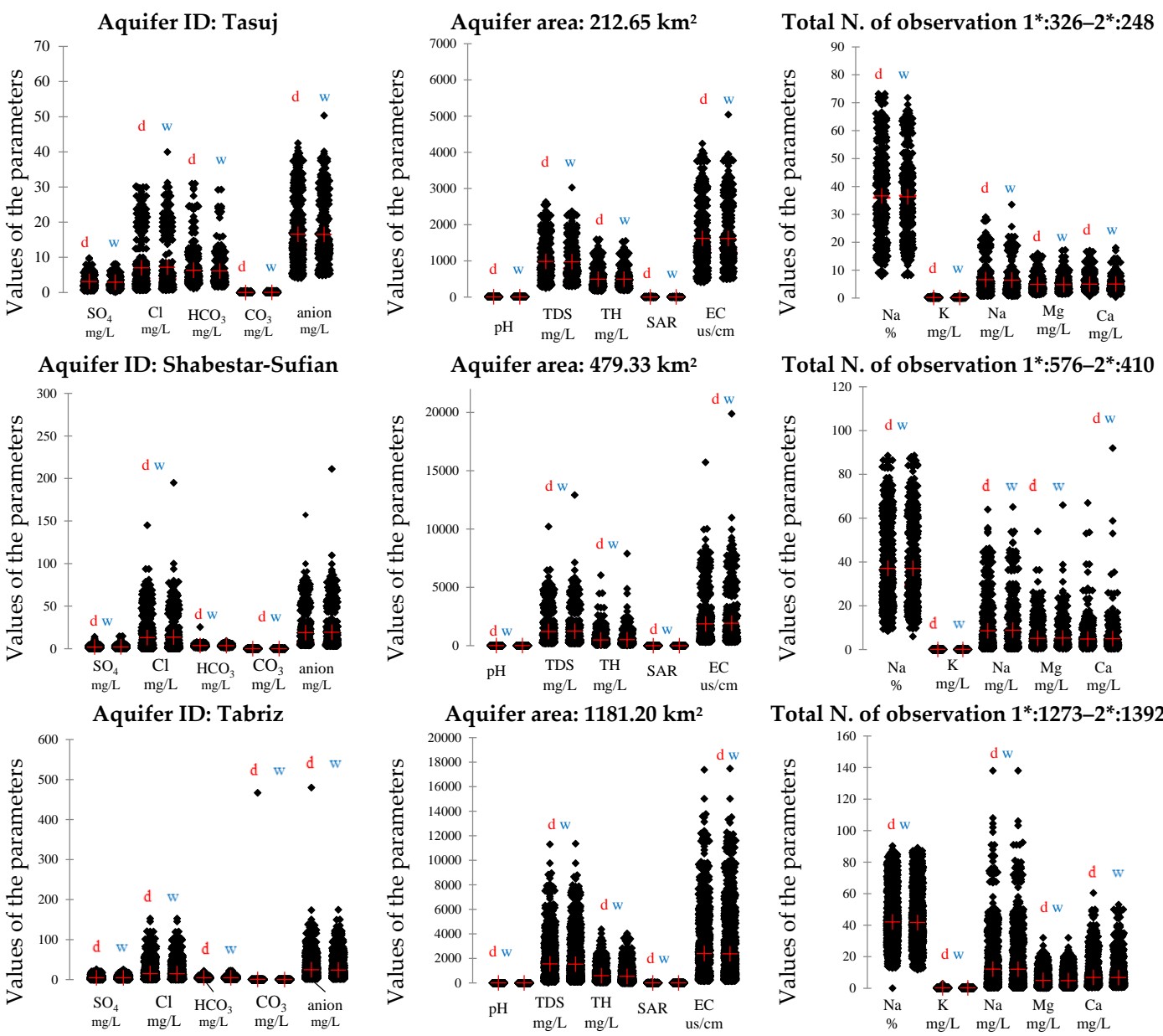

**Figure 5.** *Cont.*

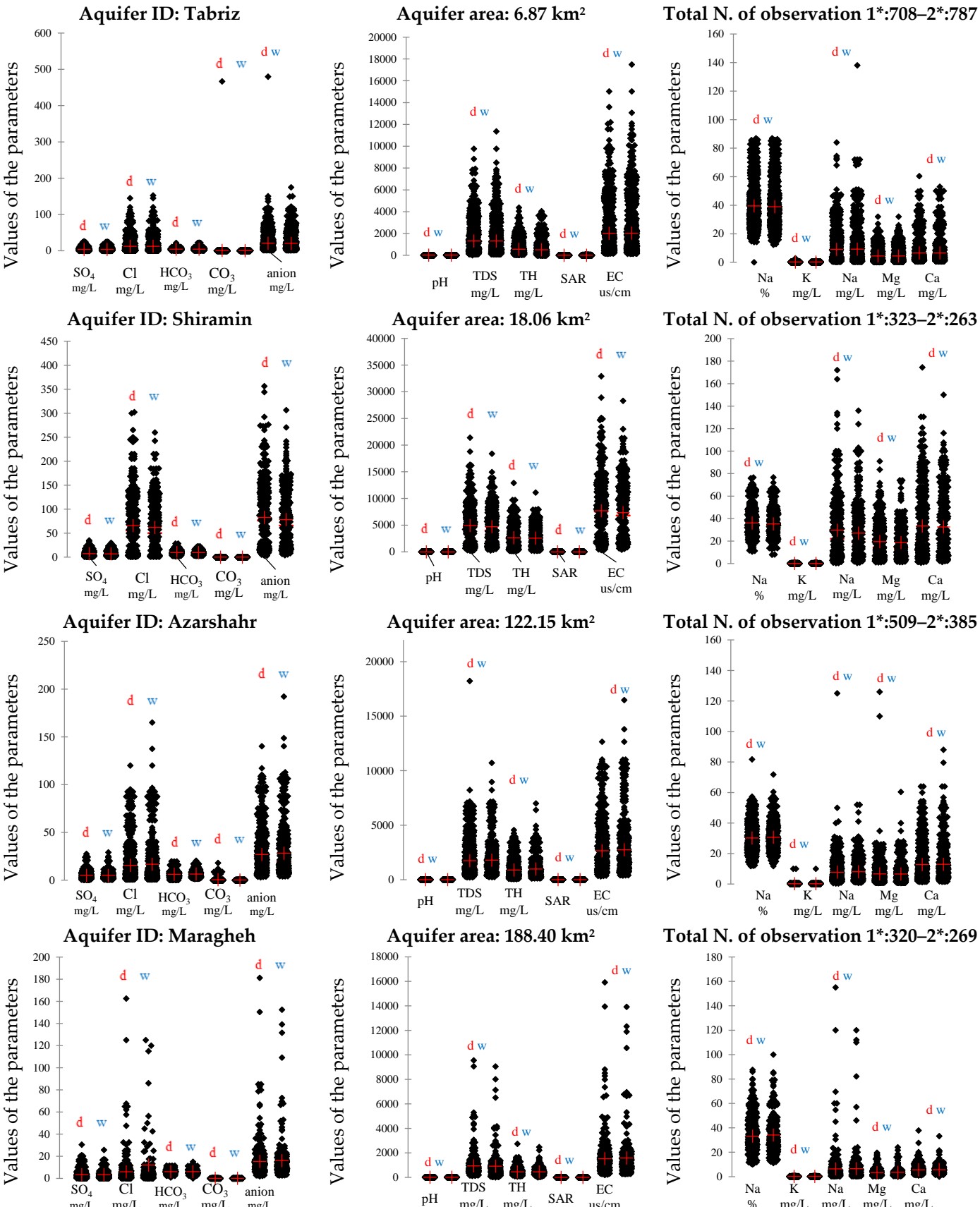

**Figure 5.** *Cont.*

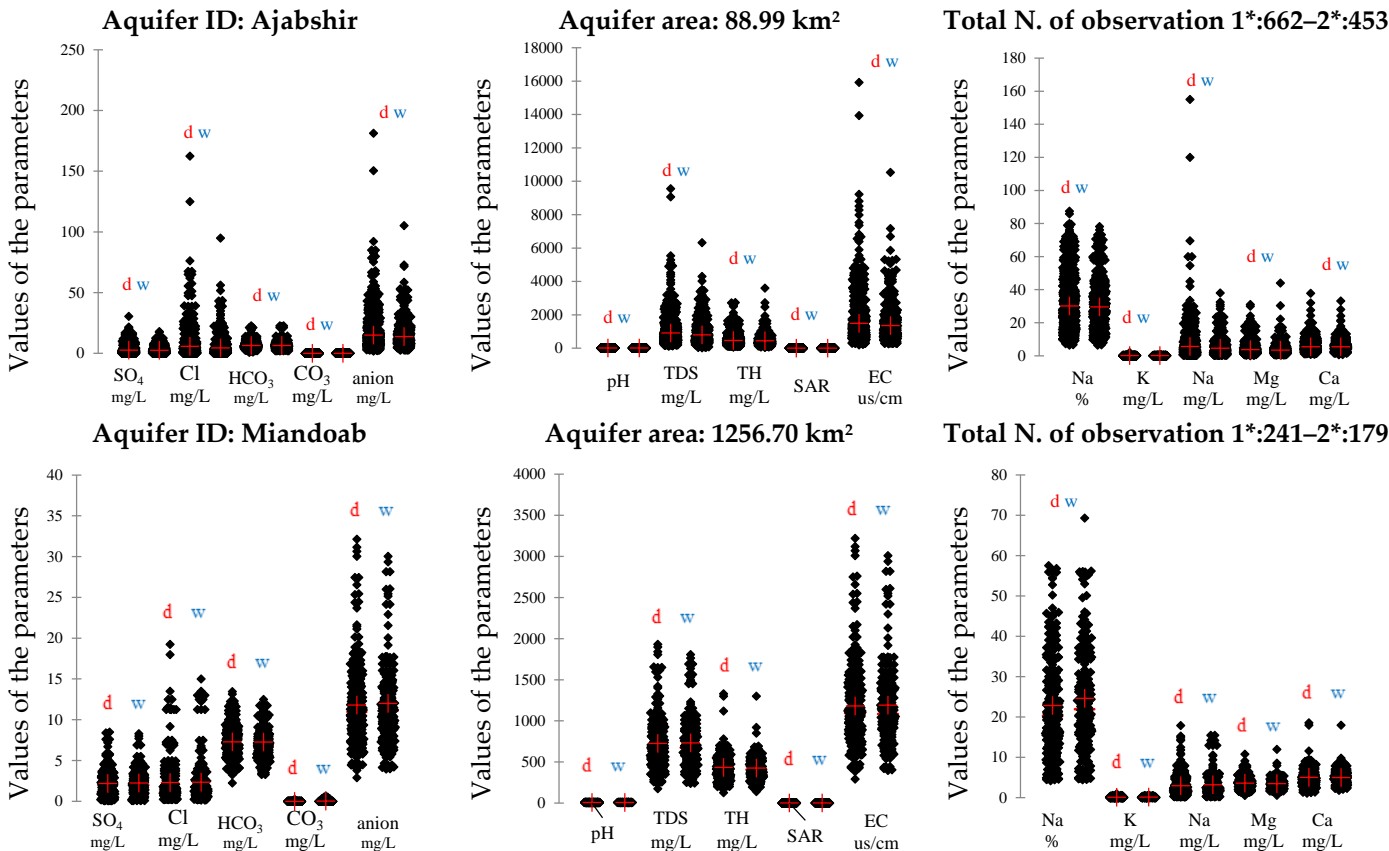

**Figure 5.** Statistical analysis of the chemical concentrations in the groundwater from nine aquifers on the eastern shore of Lake Urmia. The notation "d" and "w" refer to the dry and rainy seasons, respectively. Note: SAR, sodium adsorption ratio; TH, total hardness; TDS, total dissolved solids; EC, conductivity.

## 4. Results

### 4.1. Groundwater Quality

Descriptive statistics for the hydro-chemical parameters of the groundwater samples collected for the 20-year period from 356 wells, springs, and qanats in the dry and wet seasons are presented in Figure 5. As indicated in the results, the major cations in the groundwater database had the following order of concentration: $Na^+ > Ca^{2+} > Mg^{2+} > K^+$ in the Tasuj, Tabriz (a), Tabriz (b), and Maragheh aquifers during both seasons; $Na^+ > Mg^{2+} > Ca^{2+} > K^+$ in the Shabestar-Sufian aquifer during both seasons; $Ca^{2+} > Na^+ > Mg^{2+} > K^+$ in the Shiramin and Azarshahr aquifers during both seasons; $Na^+$ (dry season) > $Ca^{2+} > Na^+$ (wet season) > $Mg^{2+} > K^+$ in the Ajabshir aquifer; and $Ca^{2+} > Mg^{2+} > Na^+ > K^+$ in the Miandoab aquifer during both seasons. The major anions were detected in the following order of concentration: $Cl^- > HCO_3^- > SO_4^{2-}$ in the Tasuj, Tabriz (a), Tabriz (b), Shabestar-Sufian, Shiramin, and Azarshahr aquifers during both seasons; $Cl^-$ (wet season) > $HCO_3^- > Cl^-$ (dry season) > $SO_4^{2-}$ in the Maragheh aquifer; and $HCO_3^- > Cl^- > SO_4^{2-}$ in the Ajabshir and Miandoab aquifers during both seasons. As can be seen in Figure 5, Na+ was the dominant cation (except in the Azarsahr and Miandoab aquifers) and $Cl^-$ was the dominant anion (except in the Miandoab aquifer) in the major ion chemistry of the groundwater in all aquifers during both seasons.

The total dissolved solids (TDS) and conductivity (EC) values showed an increasing trend from the north (the Tasuj aquifer had 10-year dry/wet season mean values of 985.914/977.191 mg/L and 1620.16/1615.98 µS/cm, respectively) to the middle part of the basin with dry/wet season maximum values of 4923.971/4712.58 mg/L and 7696.149/7344.586 µS/cm, respectively, for the Shiramin aquifer. However, a decreasing trend was observed from

the Shiramin aquifer to the south of the basin with dry/wet season minimum TDS and EC values of 726.578/731.158 mg/L and 1182.498/1191.335 µS/cm, respectively, in the Miandoab aquifer.

### 4.2. Groundwater Quality Analysis

WQI values were computed to assess groundwater quality for drinking purposes in the study area. The average WQI values obtained from the groundwater samples for the two seasons (n = 712) are presented in Figure 6. According to the literature, e.g., [58], the WQI values of potable water are categorized into five water quality classes (Table 1). Our results indicate that in the study area, the groundwater at 199 and 178 sites was good quality for drinking purposes according to WHO standards during the dry and wet seasons, respectively. In contrast, the groundwater at 159 and 176 sites was poor and very poor quality during the dry and wet seasons, respectively. Furthermore, some of the water samples were found to be entirely unsuitable for drinking purposes. Figures 7 and 8 show the groundwater quality index maps for the dry and rainy seasons during the study period, where blue denotes low WQI values and good groundwater quality. The WQI values in the dry season ranged between 31 and 1339.1, with a spatial mean of 152.76. The northern (Tasuj aquifer) and southern (Maragheh and Miandoab aquifers) parts of the study area showed low values, which correspond with relatively good groundwater quality. However, the central parts (Shiramin and Tabriz aquifers) had high WQI values, which indicate poor water quality. The WQI values in the wet season varied between 33.16 and 1024.686, with a spatial mean of 152.87.

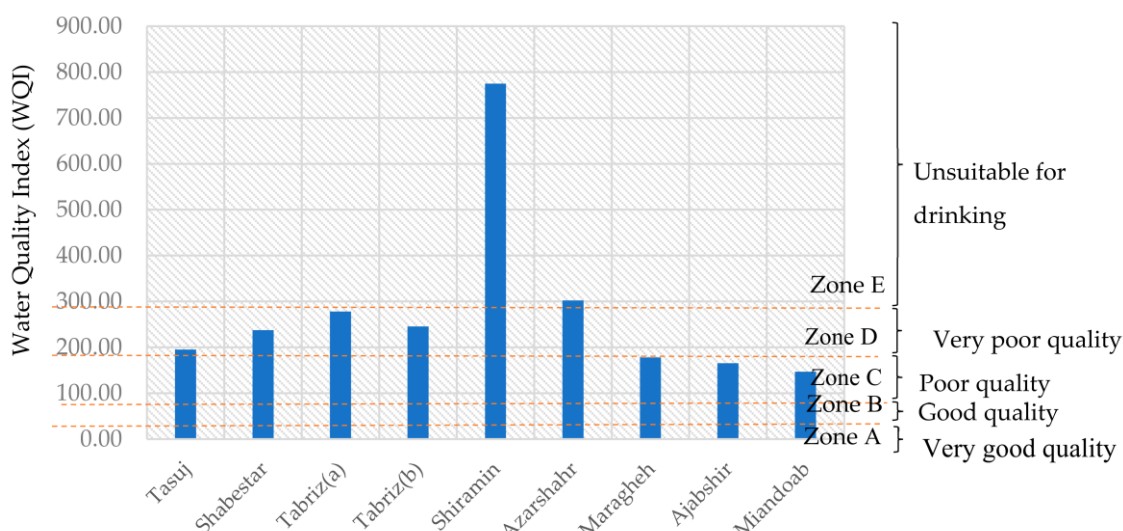

**Figure 6.** Water quality status in the aquifers of the study area.

Although the WQI indicated poor quality for both seasons, spatial variability in WQI values is likely to be connected to land-use activities [59]. When comparing the overall groundwater quality between the dry and wet seasons, a slight decline in mean WQI was detected from the wet to the dry season, indicating that the overall groundwater quality had decreased slightly from the dry to the wet season (Table 2). This could be due to seasonal recharge when precipitations washed away material deposited in the vadose zone (underground water above the water table) and into the groundwater zone. The same findings were reported by Khan et al. [59]. Table 3 lists the correlation matrix between the groundwater quality indexes calculated for each aquifer and the quality variables. A correlation coefficient of $r \geq 0.75$ is considered significant (italics in Table 3). Total hardness, total dissolved solids, and salinity had a strong positive relationship (>78.5%, >80.3%, and >70.6%) with WQI for all aquifers, whereas pH, $HCO_3^-$, and $CO_3^{2-}$ did not show a strong relationship with WQI in the study area. $SO_4^{2-}$ and $Cl^-$ were similar to salinity in that

they also had a positive relationship (>78.2%) for most aquifers. $CO_3^{2-}$ and pH showed a slight negative correlation with WQI.

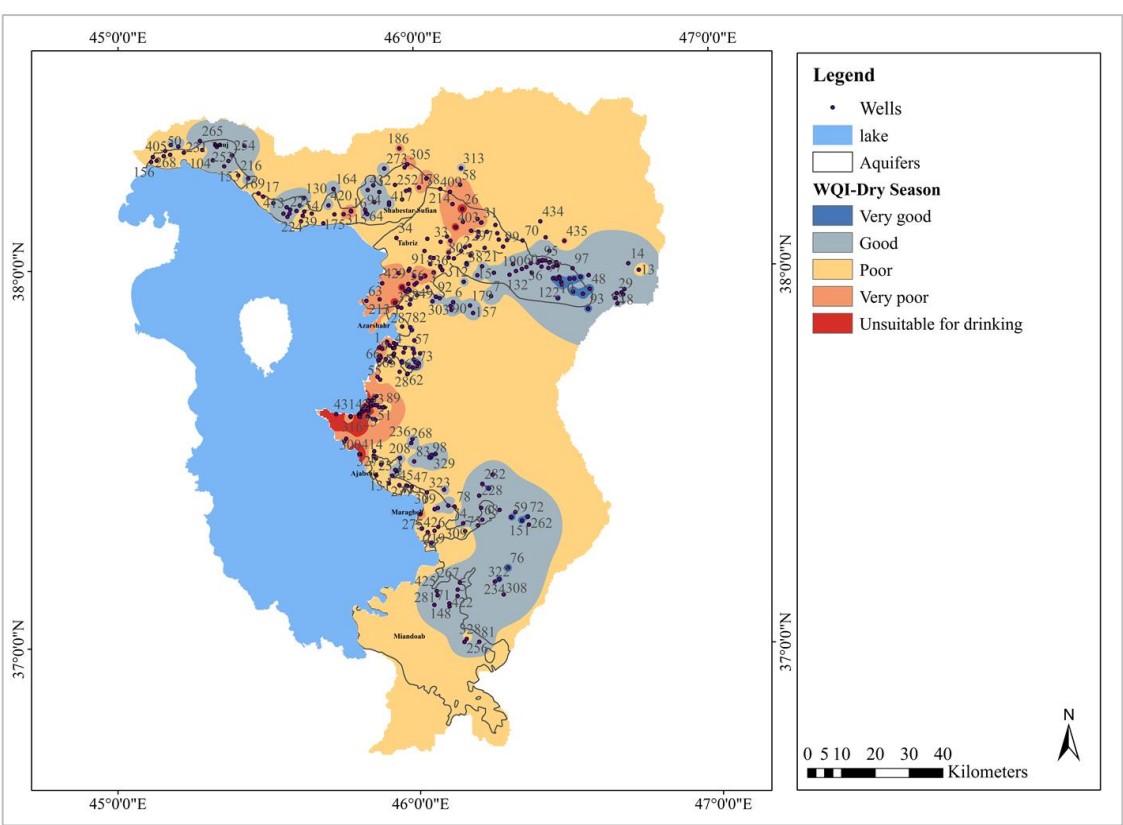

**Figure 7.** Water Quality Index map during the dry season.

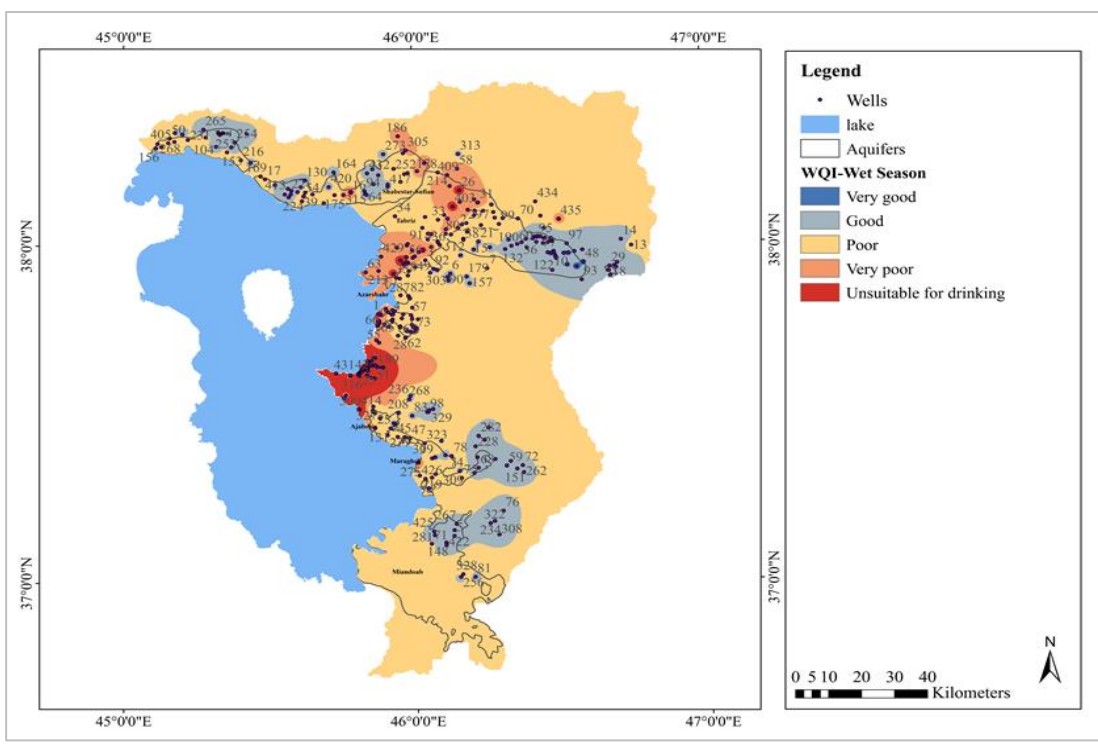

**Figure 8.** Water Quality Index map during the wet season.

**Table 2.** Assessment of the quality of groundwater for drinking purposes based on the WQI. Number of observations: 358 observations in the dry season and 354 observations in the wet season.

| | No. of Observations | | | |
|---|---|---|---|---|
| **Aquifer** | **Good to Very Good Quality Classes** | | **Poor to Unsuitable for Drinking Classes** | |
| | **Dry Season** | **Rainy Season** | **Dry Season** | **Rainy Season** |
| Tasuj | 13 | 12 | 8 | 9 |
| Shabestar | 24 | 24 | 17 | 17 |
| Tabriz (a) | 65 | 59 | 63 | 69 |
| Tabriz (b) | 7 | 7 | 1 | 1 |
| Shiramin | 9 | 1 | 29 | 37 |
| Azarshahr | 12 | 8 | 17 | 21 |
| Maragheh | 19 | 19 | 6 | 6 |
| Ajabshir | 34 | 34 | 14 | 14 |
| MiandoAb | 16 | 14 | 1 | 2 |
| Total | 199 | 178 | 159 | 176 |

**Table 3.** Spearman correlation results between WQI and variables.

| Variables | Tasuj | Shabestar | Tabriz (a) | Tabriz (b) | Shiramin | Azarshahr | Maragheh | Ajabshir | MiandoAb |
|---|---|---|---|---|---|---|---|---|---|
| $SO_4^{2-}$ | 0.427 | **0.850** | **0.910** | **0.768** | **0.839** | 0.652 | **0.817** | **0.850** | **0.782** |
| $Cl^-$ | **0.836** | **0.968** | **0.964** | 0.685 | **0.909** | **0.916** | **0.937** | **0.937** | **0.868** |
| $HCO_3^-$ | 0.273 | 0.559 | 0.636 | 0.565 | 0.205 | 0.491 | 0.522 | 0.499 | **0.780** |
| $CO_3^{2-}$ | −0.401 | −0.464 | −0.573 | −0.144 | −0.268 | −0.126 | −0.077 | −0.046 | 0.200 |
| anion | **0.960** | **0.960** | **0.969** | **0.832** | **0.911** | **0.952** | **0.968** | **0.956** | **0.952** |
| pH | −0.384 | −0.620 | −0.459 | −0.402 | −0.465 | −0.362 | 0.193 | 0.048 | 0.150 |
| TDS | **0.971** | **0.958** | **0.977** | **0.803** | **0.912** | **0.955** | **0.960** | **0.948** | **0.946** |
| TH | **0.822** | **0.881** | **0.924** | **0.785** | **0.881** | **0.948** | **0.909** | **0.892** | **0.788** |
| EC | **0.971** | **0.960** | **0.803** | 0.606 | 0.660 | 0.696 | **0.853** | **0.876** | **0.908** |
| SAR | 0.706 | **0.866** | **0.976** | **0.859** | **0.908** | **0.956** | **0.968** | **0.956** | **0.947** |
| $K^+$ | 0.513 | **0.866** | **0.805** | 0.503 | 0.475 | 0.719 | 0.606 | 0.551 | **0.796** |
| $Na^+$ | **0.846** | **0.931** | **0.954** | **0.797** | **0.800** | **0.887** | **0.931** | **0.935** | **0.950** |
| $Mg^{2+}$ | 0.747 | **0.914** | **0.946** | **0.781** | **0.901** | **0.921** | **0.908** | **0.906** | **0.834** |
| $Ca^{2+}$ | **0.830** | **0.826** | **0.835** | 0.671 | **0.833** | **0.889** | **0.750** | 0.748 | 0.586 |
| WQI | 1 | 1 | 1 | 1 | 1 | 1 | 1 | 1 | 1 |

Note: SAR, sodium adsorption ratio; TH, total hardness; WQI, Water Quality Index.

Figure 9 shows a summary of the PCA analysis of 356 wells, 9 aquifers, and 2 dry and wet seasons during 20 years with 14 quality variables. From the PCA, we extracted two principal components for the Tasuj, Shabestar, Tabriz (a), Azarshahr, Maragheh, Ajabshir, and Miandoab aquifers, and three principal components for the Tabriz (b) and Shiramin aquifers, with eigenvalues greater than one, to explain over 80% of the variations. The results yielded component scores for groundwater quality variables and observations, representing the contribution each variable to the two main components. The size of the observations was changed based on their squared cosines to reflect the representation quality of each well in each season on the PCA axis. The small sizes of the squared cosines associated with an axis ($cos2 < 0.4$) have not been interpreted.

### 4.2.1. Tasuj Aquifer

The PCA produced a correlation matrix between all variables and observations (Figure 9). In the Tasuj aquifer, a high positive correlation coefficient was observed between $Cl^-$ and $Na^+$ ($r = 0.939$), and $Mg^{2+}$ and $Ca^{2+}$ ($r = 0.843$). Furthermore, high positive correlations between TDS and EC ($r = 0.998$), $Na^+$ ($r = 0.83$), $Mg^{2+}$ ($r = 0.816$), $Ca^{2+}$ (0.867), and TH ($r = 0.887$) were measured as expected. The correlations for pH and $CO_3^{2-}$ were negative for almost all the major elements, which reflects the hydro-chemical attributes of the aquifer [60].

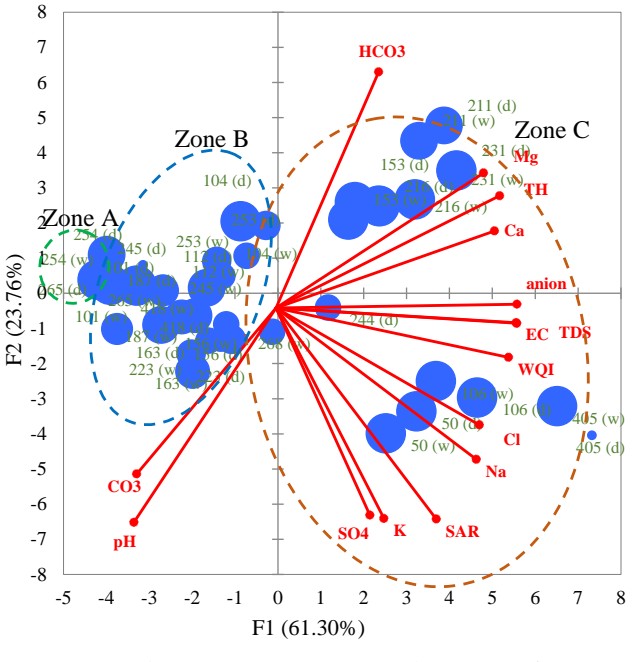

Observations (axes F1 and F2: 85.06%)
*Tasuj*

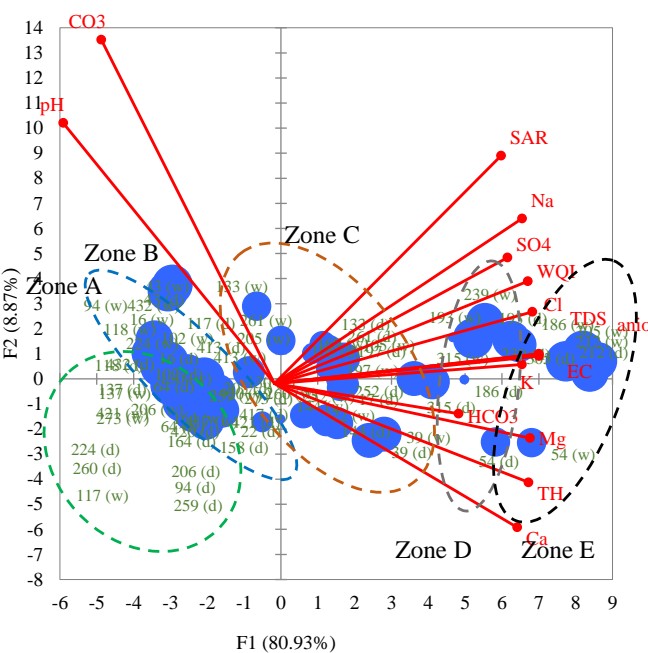

Observations (axes F1 and F2: 89.80%)
*Shabestar*

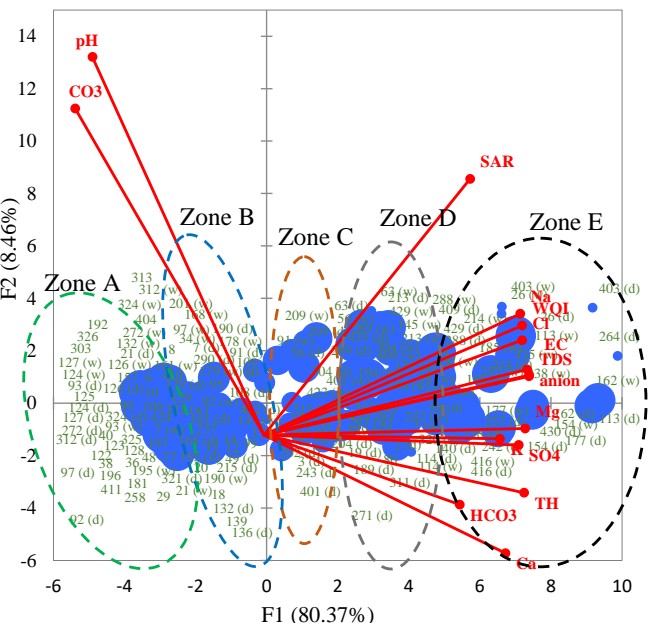

Observations (axes F1 and F2: 88.83%)
*Tabriz (a)*

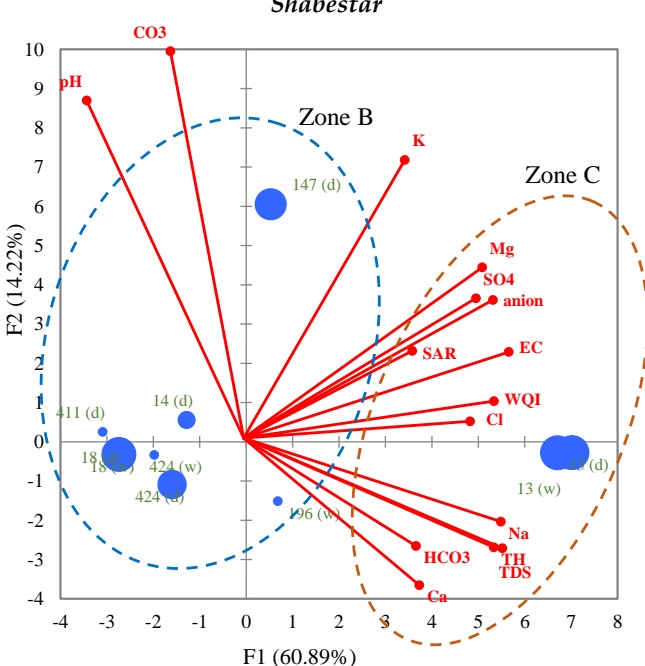

Observations (axes F1 and F2: 75.11%)
*Tabriz (b)*

**Figure 9.** *Cont.*

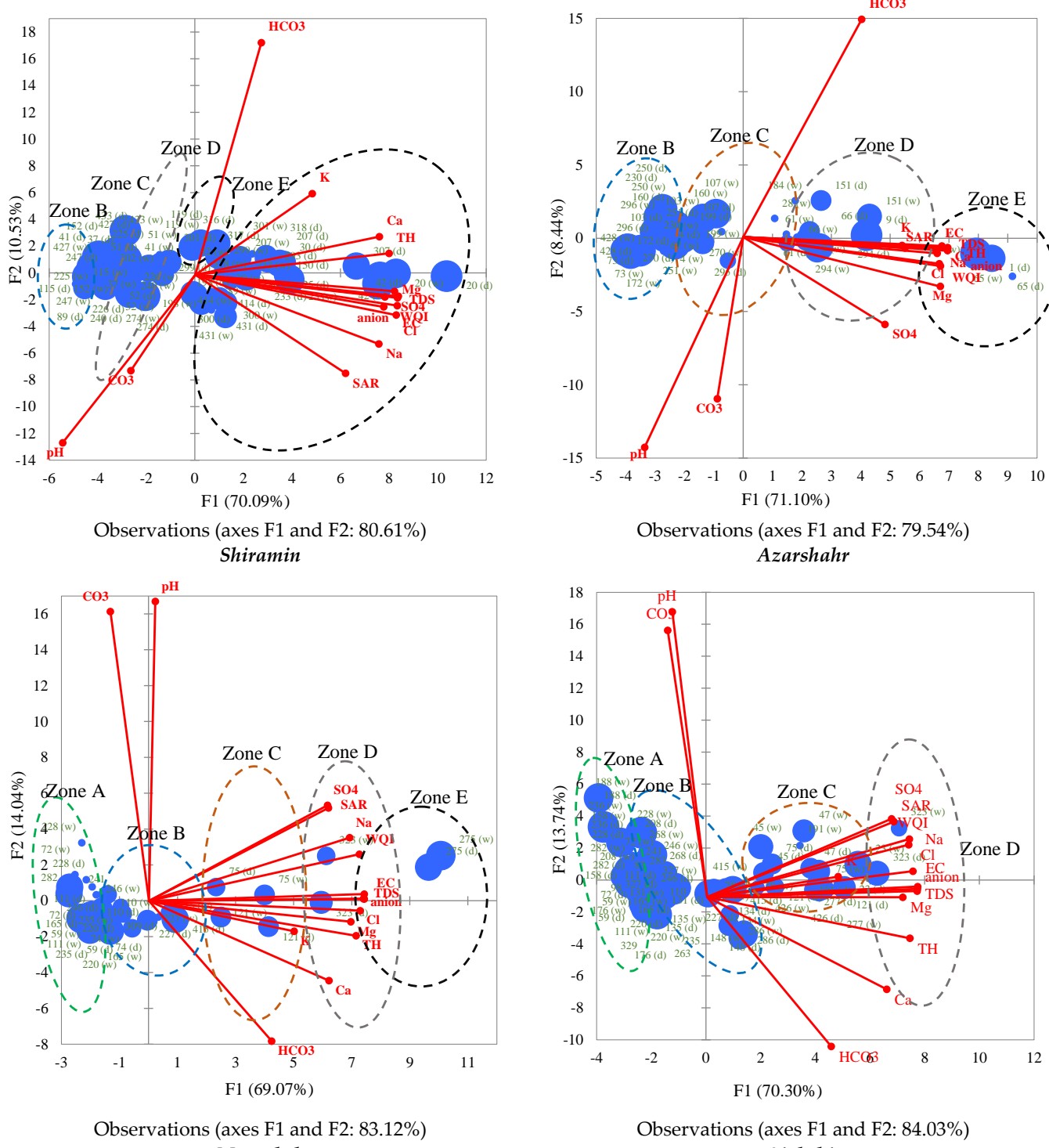

**Figure 9.** *Cont.*

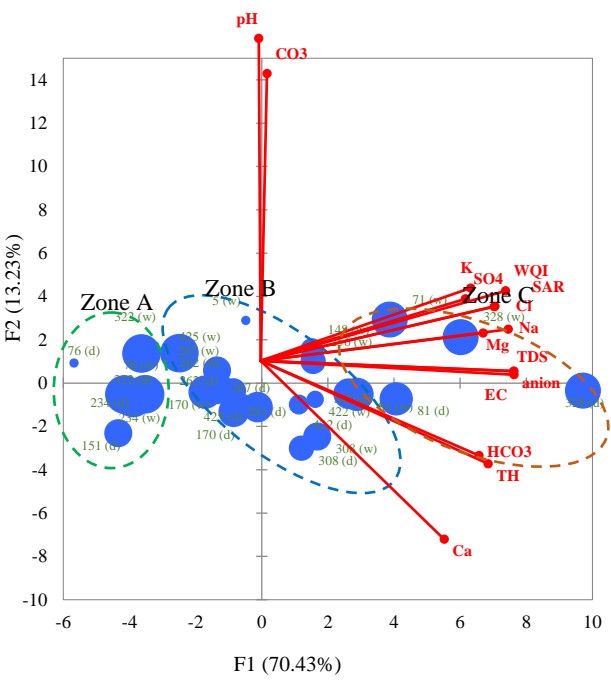

Observations (axes F1 and F2: 83.66%)
*Miandoab*

**Figure 9.** PCA analysis of 356 wells in 9 aquifers and 2 dry (d) and rainy (w) seasons during the 20-year study period with 14 quality variables combined with cluster analysis results (Zone A to E).

According to our results, factor F1 explained 61.3% of the variance. It had a positive correlation with the original variables of $Cl^-$, anions, TDS, TH, EC, $Na^+$, $Mg^{2+}$, $Ca^{2+}$, and WQI (the contribution of these variables were over 82% in the new factor F1). High factor scores indicate high SAR, $Na^+$, $Mg^{2+}$, $Ca^{2+}$, $HCO_3^-$, $SO_4^{2-}$, $K^+$ concentrations, and high EC, TDS, and TH values. According to the results, approximately 33% of the groundwater samples were slightly saline (1000 < TDS < 3000 mg/L) based on groundwater quality classification [17]. All TDS values were above 2100 mg/L, indicating the influence of the local geology on groundwater quality. The total concentration of dissolved calcium and magnesium in the samples was linked to hardness. In terms of hardness, water is categorized as soft, hard, moderately hard, and very hard. The majority (83%) of the samples taken in our study area were very hard (TH > 300). The results for the WQI and clustering analysis classified the groundwater quality of this aquifer into the three groups, viz. very good (Zone A), good (Zone B), and poor (Zone C) quality (Figure 9). The factor score for Zone A was F1 < −4, where groundwater quality was identified as being very good. The qanat with the code "254" in Zone A was stratigraphically located in Plqc (fluvial conglomerate, piedmont conglomerate, and sandstone) and was a pasture land-use type. On the other hand, the deep well "405" with F1 > 7 and WQI > 190, which was geologically located in URig (red marl, gypsiferous marl, sandstone, and conglomerate) and was an agricultural-rainfed land-use type, was classified as Zone C.

### 4.2.2. Shabestar-Sufian Aquifer

In this region, the strong positive loadings for $K^+$ and $Cl^-$ (F1 > 0.9) were interpreted as diffused forms of contamination resulting from the application of chemical fertilizer such as NPK, Potash, and manure [61]. Five hydro-chemical groundwater zones were identified based on the distribution map of factor scores for the 41 observation wells in the Shabestar aquifer (Figure 9). Zone A (F1 < −1), with a low mineralization, included wells 164 (in both seasons) in bare lands (land-use type) and flysch turbidites (K2ft) (geologically); wells 206, 224, 260, and 273 (in the dry season) in agricultural and rangeland areas (land-use types); and Qt1 (high-level piedmont fan and valley terrace deposits) (geologically). The

results for the WQI also confirmed the PCA outcomes. Wells 43 (dry and wet seasons) and 133 (wet season) showed a high correlation with F2. The groundwater quality in these wells was good (Zone B) with low mineralization and high pH (pH > 8), and $CO_3^{2-}$ ranging between 0.3 and 0.4 mg/L, which was related to the farmlands. Zone E (wells 54, 212, 305 in both seasons) contained the most mineralized groundwater with F1 > 6.8. Approximately 31% and 12% of the samples were recognized as slightly saline (1000 < TDS < 3000) and moderately saline (3000 < TDS < 10000), respectively [17]. Wells with the codes 186, 193, 239, and 315, located in Qt1 and URM (light red to brown marl and gypsiferous marl), had highly mineralized groundwater. These wells, which are next to urban and agricultural areas, were classified as very poor water quality (Zone D).

### 4.2.3. Tabriz Aquifer

Based on the results, almost all quality variables (except pH) yielded a positive correlation with factor F1 (80.37% of the variability) for the Tabriz (a) aquifer. $CO_3^{2-}$ and pH correlated better with F2, whereas $HCO_3^-$ was 0.64 in the component F3. A closer look at Figure 9 reveals that the groundwater in the wells (26, 113, 154, 162, 177, 185, 213 (wet season), 214, 242 (wet season), 257, 264, 288 (wet season), 320, 403, 429, 430 (wet season) and 435 (wet season)) with factor scores F1 > 4.5 were highly mineralized with TDS values between 3172.4875 and 7525.375 mg/L and total hardness ranging between 1271 and 1444.33 mg/L. Approximately 81% and 58% of the samples were moderately saline (3000 < TDS < 10,000) and very hard (TH > 300); therefore, based on the groundwater classification, they are unsuitable for potable supply [17]. The $Ca^{2+}$ concentration exceeded $Mg^{2+}$ concentration at many sites for this aquifer, which is indicative of limestone and the significant presence of calcium-bearing minerals in sedimentary rocks (Figure 5). According to the results of the cluster analysis, the groundwater quality for the Tabriz (a) and Tabriz (b) aquifers can be classified into five (Zone A to E) and two (Zone B and C) classes, respectively (Figure 9).

### 4.2.4. Shiramin Aquifer

The wells with high factor scores (wells 20, 25, 30, 42, 89 (wet season), 150, 207, 299, 302 (wet season), 307, and 318) were classified into Zone E (unsuitable as drinking water). Approximately 7% and 98% of the samples were very saline (10,000 < TDS < 30,000) and very hard (TH > 300), respectively, based on the groundwater classification. The high ratio of calcium and magnesium in the groundwater samples is an indicator of the hardness in this aquifer (Figure 5). Compared to the other studied aquifers, the highest concentrations of $Ca^{2+}$ (with maximum and average values of 136.05 and 35.8 mg/L, respectively) and $Mg^{2+}$ (with maximum and average values of 66.05 and 20.54 mg/L, respectively) were found in Shiramin. In geological terms (Figure 4), the Shiramin basin is mostly composed of thick-bedded orbitolina limestone, tile red sandstone, and gypsiferous marl. The minerals contributing to the hardness of the water enter the groundwater as it percolates through materials containing calcium and magnesium (e.g., limestone, gypsiferous marl, and sedimentary rocks). Hard water can cause scaly deposits inside pipes and tanks and may cause metal corrosion. It is thus not desirable for domestic use. Furthermore, the dissolved salts of primarily calcium and magnesium found in hard water have several effects on human health and can lead to urolithiasis, anencephaly, prenatal mortality, some types of cancer, and cardiovascular diseases [62]. The chloride concentration was higher than the $HCO_3^-$ concentration (see Figure 5), which indicates the influence of lake water ingression. The higher chloride content may be due to the weathering of rocks, chemical fertilizers, and industrial effluent in sewage. The wells between zone B and D in Figure 9 had an intermediate chemical composition.

### 4.2.5. Azarshahr Aquifer

The variables pH and $CO_3^{2-}$ showed slight negative correlations with all the elements in this aquifer. The results for the WQI confirmed that the groundwater in these wells,

which are in cultivated areas, is unsuitable for drinking. By contrast, wells 67 (dry season), 73, 103, 160, 172, 199 and 230 (dry season), 250, 251, 270 (dry season), 296, and 428 with factor scores F1 < −1 were classified into Zone B where the groundwater quality was good.

### 4.2.6. Maragheh Aquifer

The elements $Cl^-$, $Na^+$, $Mg^{2+}$, $Ca^{2+}$, TDS, TH, and EC had very similar values (r > 0.85) for the Maragheh aquifer, which indicates that there were similar trends in their changes. On the other hand, samples from the semi-deep wells 75, 275, and 323 (wet and dry seasons), all of which were in cultivated areas, had a high positive correlation with F1. The quality of these wells is thus mainly characterized by $Cl^-$, $Na^+$, $Mg^{2+}$, SAR, $Ca^{2+}$, TDS, TH, and EC, whereas the wet season samples showed a stronger correlation with SAR and $Na^+$ for all mentioned wells. This may be because small quantities of salt accumulate in the surface layers of the soil after each rain event or with irrigation of arable lands. Evaporation and plant transpiration also remove water from the soil, leaving salt behind in the dry season [34]. Concentrated salts are mobilized through aquifers over time after precipitation and drainage during the wet season [63]. Well 227, situated in garden land-use type, had a better correlation with F2 and was characterized by high $CO_3^{2-}$ and pH values. The association of these elements with this factor may be attributed to the leaching of bedrock materials [61]. This well lies within Quaternary deposits (high-level piedmont fan and terrace deposits)

### 4.2.7. Ajabshir Aquifer

The samples from the semi-deep wells 23, 47, 191, 277 (in the wet and dry seasons), 75 (wet season), 121 (dry season), and 426, which all lie within cultivated areas had a high positive correlation with F1 (factor loading > 0.5). The groundwater quality in the mentioned wells is thus mainly characterized by $Cl^-$, $Na^+$, $Mg^{2+}$, SAR, $Ca^{2+}$, TDS, TH, and EC. A stronger correlation with SAR and $Na^+$ was also observed for samples from the wet season for the mentioned wells [63].

### 4.2.8. Miandoab Aquifer

The samples from the semi-deep wells 328 (both seasons), 71 (wet season), 81, and 151 (dry season), qanats 76 and 322, and spring 234 (both seasons) made a good contribution to F1, which means that they were more influenced by $SO_4^{2-}$, $Cl^-$, anions, TDS, TH, SAR, EC, $Na^+$, Mg, and $HCO_3^-$. A similar case was reported for the Jezireh Basin, Syria [59]. High factor scores indicated high $K^+$, $SO_4^{2-}$, SAR, $Na^+$, $Mg^{2+}$, $HCO_3^-$, $Ca^{2+}$ concentrations and high EC, TDS, and TH values. The wells, qanats, and springs with a factor score less than −3.5 (in zone A) were the least mineralized in the Miandoab aquifer. According to the WQI results, the groundwater in Zone A is very good quality freshwater with TDS values between 278.93 and 448.45 mg/L and a total hardness between 206.9 and 328.1 mg/L. Zone C (wells 71, 81, and 328 in both seasons) had the most mineralized groundwater (492.5 < TH < 617.7 mg/L) with F1 > 2.5. Geologically, these wells lie within Qt1 (high-level piedmont fan and valley terrace deposits).

## 5. Discussion

Our research assessed the spatiotemporal hydro-chemical characteristics of the nine aquifers around Urmia Lake. The results revealed that for the studied aquifers, $Na^+$ was the most prevalent cation and $Cl^-$ was the dominant anion in the groundwater during both the dry and wet seasons. According to our results, the main cations in Urmia Lake water were $Na^+$, $K^+$, $Ca^{2+}$, and $Mg^{2+}$, whereas $Cl^-$, $SO_4^{2-}$, $HCO_3^-$ were the main anions in the lake. It is worth mentioning that the $Na^+$ and $Cl^-$ concentrations in Urmia Lake are four times higher than natural seawater [12], reinforcing the hypothesis that saltwater migrates into the freshwater aquifers under the influence of hydraulic gradients toward the pumped wells. As more groundwater is extracted, more saltwater intrudes into the aquifer (e.g., higher $Na^+$ and $Cl^-$ concentrations were detected in the Tabriz aquifer, which

supplies a region with a higher groundwater demand). When comparing the dry and wet seasons, the $Na^+$ content was slightly higher during the dry season. Water demand increases during the dry season, during which aquifer recharge is minimal due to the lack of precipitation, resulting in an increased rate of saltwater intrusion into the aquifer. Furthermore, saltwater intrusion was higher in the southern aquifers (the shallower part of the lake has a higher net evaporation rate compared to the north of the lake).

As one of the major anions, chloride originates from the dissociation of salts such as sodium chloride or calcium chloride and is a good indicator of salinity [64]. Thus, the high concentration of chloride in groundwater samples is definitive proof that the saltwater of the lake affects the quality of the aquifers. The total dissolved solids (TDS) and conductivity (EC) values showed an increasing trend from the north to the middle part of the basin with the maximum values for the Shiramin aquifer. The EC is explained by hydro-chemical processes during water-rock interactions. High TDS levels can be caused by the discharge of municipal and industrial effluents, as well as industrial seepage and the percolation of solids-laden channel water. Thus, salinization can be caused by both natural and anthropogenic processes, although they are often intertwined. In all the studied aquifers (except the Miandoab aquifer), the chloride concentration was higher than the $HCO_3$ concentration, indicating that mineral dissolution had occurred in the study area.

We employed the PCA to reduce fourteen quality variables to two primary components for the Tasuj, Shabestar, Tabriz (a), Azarshahr, Maragheh, Ajabshir ,and Miandoab aquifers and three primary components for the Shiramin and Azarshahr aquifers that could explain over 80% of the total variance in the original dataset. Factor F1 (F1 > 60%) represented groundwater mineralization from $SO_4^{2-}$, $Cl^-$, anion, TDS, TH, SAR, EC, $K^+$, $Na^+$, $Mg^{2+}$, and $Ca^{2+}$, whereas F2 (F2 > 8.4%) represented groundwater mineralization from $HCO_3^-$, $CO_3^{2-}$, and pH, all of which affect groundwater quality in the ULB. The association of $HCO_3^-$, $CO_3^{2-}$, and pH with factor F2 may be attributed to the leaching of bedrock materials and rock-water interaction. The $Mg^{2+}$ content in groundwater is mostly controlled by rock weathering [27]. A strong positive correlation for $K^+$ (F1 > 0.9) was found in the Shabestar and Tabriz (a) aquifers and was interpreted as the result of contamination caused by the application of chemical fertilizers such as NPK, potash, and manure. $K^+$ in groundwater may also result from the interaction between water and silicates [65].

Based on the WQI values computed to assess groundwater quality in the study area for potable supplies, approximately 48% of the groundwater samples were identified as poor to unsuitable for drinking purposes according to WHO standards. Total hardness, total dissolved solids, and salinity had a strong positive relationship (>78.5%, >80.3%, and >70.6%, respectively) with the WQI in all aquifers, whereas pH, $HCO_3^-$, and $CO_3^{2-}$ did not show a strong relationship with the WQI in the study area. $SO_4^{2-}$ and $Cl^-$ also had a positive correlation (>78.2%) in most aquifers. $CO_3^{2-}$ and pH showed a slight negative correlation with the WQI. While the WQI spatial means for both seasons denoted relatively poor was quality, the spatial variability in the WQI values may be linked to land-use activities. Comparing the overall groundwater quality between the dry and wet seasons showed a slight decrease in the mean WQI value, which indicated that overall groundwater quality had slightly decreased from the dry to the wet season.

Prior studies regarding the Urmia Lake drought mentioned an increase in orchards and farmlands around the lake [14,15,66,67]. Based on the results, the extraction of groundwater from the surrounding aquifers has altered the groundwater level balance and increased the interaction of salty and fresh groundwater. The hyper-salinity of the lake water, as well as the processes of saltwater encroachment and intrusion, have had an impact on groundwater quality in the surrounding area. Excessive aquifer discharge and disruption of groundwater supplies can lower the interphase threshold, resulting in saltwater intrusion into nearby aquifers, as earlier studies have shown [34]. It is well understood that the prevalent agricultural systems in the ULB are small-scale family farms with surface irrigation systems that pump their water supply from groundwater, which accordingly, increases the pressure on aquifers [34]. Such traditional farms often produce crops with high water demands, such

as onions, tomatoes, potatoes, melons, watermelons, cucumbers, sugar beets, carrots, alfalfa, garlic, wheat, barley, grain maize, sunflowers, soybeans, beans, safflowers, canola, forage sorghum, and major horticultural crops including apples, grapes, plums, apricots, peaches, nectarines, pears, walnuts, almonds, and sour cherries. These plants and crops all require a significant amount of water, which is currently supplied from dams and groundwater. Due to a lack of precipitation from June to October, these crops depend upon irrigation networks connected to 76 dams and the groundwater discharge from 8742 wells. Thus, groundwater salinization and land subsidence are tangible environmental effects resulting from intensive pumping and groundwater extraction. Low water demand crop optimization might be an efficient approach to reduce the amount of water extracted from aquifers.

## 6. Conclusions and Outlook

Given the increasing dependence on groundwater globally, it is necessary to understand the various forms of degradation threatening the world's groundwater supplies. In this research, we examined the condition of groundwater in the aquifers of the eastern plains of Urmia Lake, which have been threatened by severe drought since 2005. The findings of this research conclude that, due to the lake drought and interaction between hyper-saline- and freshwater, the EC of groundwater, especially in the central plains, has increased significantly, which has led to it being unsuitable for potable purposes. Anthropogenic contamination sources such as chemical fertilizers, industrial waste, and untreated sewage water might be the major factors causing excessive concentrations of contaminants affecting the water quality. In addition, the balance between freshwater aquifers around the lake and the lake's hyper-saline water has been impacted by the intensive water extraction from aquifers for agricultural and drinking purposes.

From a methodological perspective, we conclude that the integrated GIS-based multivariate statistical technique and spatial analysis approach was an effective method for understanding the factors determining groundwater quality. Using the combination of PCA and WQI, we successfully related the results to the major factors, processes, and contamination sources. Considering the significance of freshwater for sustainable development, particularly in semi-arid environments (such as the ULB), the outcome of this research can be used to monitor the status of the aquifers and their hydro-chemical properties near Urmia Lake.

The results of this study demonstrated that sample wells showed relationships between nearby aquifers, which give us a new perspective on controlling the drought in this region. The geographical and geological characteristics of this mountainous region impose restrictions on finding new water resources. In this way, conservation agriculture and the use of pressurized irrigation systems are a priority for the agricultural sector in order to conserve water. As our earlier studies pointed out the intensive water scarcity in this area [68–70], our results should provide key information for decision-makers and support them in pursuing sustainable development in the northwest of Iran. The results will also be useful for mitigating the environmental impact of the Urmia Lake drought and prioritizing areas for initiatives that halt degradation and implement restoration strategies.

**Author Contributions:** Conceptualization, B.F. and Z.A.; methodology, B.F.; software, Z.A.; validation, B.F., B.S. and Z.A.; formal analysis, Z.A.; investigation, B.F.; resources, B.S.; data curation, Z.A.; writing—original draft preparation, B.F.; writing—review and editing, Z.A.; visualization, B.S.; supervision, B.F.; project administration, B.F.; funding acquisition, B.F. All authors have read and agreed to the published version of the manuscript.

**Funding:** This research was funded by University of Tabriz grant number [S4409] and Alexander von Humboldt Foundation. The APC was funded by [Humboldt-Universität zu Berlin].

**Data Availability Statement:** Data available on request.

**Conflicts of Interest:** The authors declare no conflict of interest. The funders had no role in the design of the study; in the collection, analyses, or interpretation of data; in the writing of the manuscript; or in the decision to publish the results.

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
