# Peer review of "A GIS-Based Spatiotemporal Impact Assessment of Droughts in the Hyper-Saline Urmia Lake Basin on the Hydro-Geochemical Quality of Nearby Aquifers"

_remotesensing, doi:10.3390/rs14112516_

Round 1
Reviewer 1 Report
Revision commentaries to authors (and editors).
This artcicle is focused to study of aquifers located in zones near the Urmia lake, and the water potable quality problems, using GIS and close technologies.In general terms the study is well oriented, in data adquisition and his study, using data treatment (PCA, etc.), obtaining the quality levels.
Is not very well defined the type of the data taking in the aquifers but may be direct contact with the wells.
But is very evident the necessity of investigate new sources of hidric resources to attend the potable Waters need of the people in that región.
Revision commentaries to authors only.
Comment aditional on the data taking method, with direct contact with the aquifers to take the data for the study, i think is not specifically long distance technology (remote sensing). But if the magazine may consider admissible I do not may express nothing about this aspect of the publication. Ok.
I finissed the article revision. Thank you very much.
Author Response
Dear Reviewer,
We very much appreciate of your positive statements regarding our manuscript. It appears us you kindly spent significant time on this manuscript and we are grateful for the detailed comments. In fact, you brought up interesting aspects and we believe that these comments and our respective reactions to them will improve the quality of our paper. We did our best to improve the scientific quality of the manuscript significantly. Based on your constructive comments on the early version and the comprehensive revisions on the manuscript, we are very confident that you will find this revised version now worthwhile to get published
We have revised a paper according to your comments and the other 3 reviewers. In terms of your valuable comments, we may indicate that Urmia Lake is the largest saline lake in Iran which has been drying up due to the intensive climate change, extension of farmlands and illegal withdrawals from wells by farmers. The government also constructed a large number of dams in the east section of this lake which is located in the East Azerbaijan Province is making many disasters and damage to the environment and residents of these areas. Therefore, basically due to the mismanagement intensive environmental issues have raised in this area. On the motherland the groundwater is the main resource for agricultural activities and also drinking of about 7.3 minion inhibition in this area which has been facing the salinization issue over the past years. Since this paper has submitted to the Special Issue "Recent Geospatial Methods and Techniques for Urban Water Management", therefore we aimed to analysis the trend of aquifer salinization in this area. We agree that it is mostly based on GIS teschenites rather than remote sensing, however since it is being submitted for the especial issues and covers the aims a scope of the especial issue we hope that you confirm the paper relevancy to your journal. In addition, we also improved our statements and justification and the paper’s contribution for urban water management scopes in the revised version.
We also may indicate that, in this paper, the available aquifers were accurately determined and the potential of the area and the main watersheds that strongly affected the level of Lake Urmia are considered for the first time. Moreover, the value of each well in the drought was also considered and the critical wells were determined for the first time. Considering the wells also showed the relations of nearby aquifers which give us new attitudes toward controlling the drought in this region. The geographical and geological characteristics of this mountainous region impose restrictions on finding new water resources. In this way, conservation agriculture and the use of pressurized irrigation systems are a priority for the agricultural sector in order to conserve water. Totally, like other studies, further studies and new and upgraded methods will complete our findings.
Once again, many thanks for your comments and supportive statements regarding our submission, we honestly did our best to improve the paper further and we really hope that you will agree the paper to be accepted and published.

Reviewer 2 Report
Please provide line number for the manuscript for revision.
Please avoid unnecessary (non-relevant) self-citations.
Reference entry 50: Please provide published resources.
Introduction:
Page 1: Second last line: Typo in the superscript.
Figure 1: Fix Y-axis labels
Figure 2: Figure 2A is quite confusing, could not understand what you were displaying, please provide labels. Does black bar show day? Cumulative rainfall? What is the blue line graph on the bottom?
Figure 3: Fix legend, remove underscore. What is rock “output”? You mean outcrop?
Clean-up and generalize your units. Simplify your scale bar too. Same comments for other map scales.
Rename some of the units, garden could be recreational land? Maybe merge Pasture 1 and 2.
Enlarge the labels on the map or make them stand out; currently they are difficult to read.
Figure 4: This figure is useless unless you provide the generalized name or provide the broader class for the units. Currently, the map has too many units and are too difficult to distinguish at this scale. So simply the map.
Page 5: Last line: Correct the formatting (sub/superscript) for the ionic charges. Same comment for section 3.3 and all across the manuscript.
Section 3.2: What kinds of satellite data? Provide a few details.
Section 3.3: Correct the subscript formatting for the variables. Correct those all across the manuscript where they appear, this applies to other variables too.
Section 4.2: Second paragraph: First line-grammar?
Figure 6: Anion label needs to be fixed for all the graphs. You have not explained what is “SAR”. Also is it “TH” or “Th”?. Please maintain consistency with your symbols. Please expand your caption a little bit, it is confusing to look at first. What does the red cross-hair symbol represent? Explain.
Figure 7: label Y-axis.
Table 3: What do those red values indicate? Maybe change the red color to italics or something since you cannot have a colored text on the table.
Figure 8 and 9: Legend not clear. What is “Aq cs”? Also, what does those numbers indicate? Explain in the caption.
Figure 10: The graphs need to be cleaned a little bit, difficult to read. Maybe you can change the blue filled circles to empty ones to make it more readable?
Page 20: Under Maragheh Aquifer:
Correct grammar for the first line.
Last line of the paragraph, missing period.
Discussion, page 21:
Last paragraph, 5th line, typo: space needed.
Page 22, second paragraph:
Correct the grammar for the line “SO42− and Cl− were like salinity in that they also have a positive correlation……”
In the same paragraph: last line starting with “When comparing the overall groundwater ….”
It is confusing, which season has better quality, wet or dry? The sentence is convoluted, please rephrase it.
Author Response
Dear Reviewer,
We very much appreciate of your positive statements regarding our manuscript. It appears us you kindly spent significant time on this manuscript and we are grateful for the detailed comments. In fact, you brought up interesting aspects and we believe that these comments and our respective reactions to them will improve the quality of our paper. We did our best to improve the scientific quality of the manuscript significantly. Based on your constructive comments on early version and the comprehensive revisions on the manuscript, we are very confident that you will find this revised version now worthwhile to get published.
Comments to the Author and reply:
Page 1: Second last line: Typo in the superscript.
Checked and corrected.
Figure 1: Fix Y-axis labels.
The labels are vertically shown the left side of the figure which show 37-38 N degree.
Figure 3: Fix legend, remove underscore. What is rock “output”? You mean outcrop?
Thank you! the aim was rock outcrop. The required issues checked and corrected as you kindly proposed.
Clean-up and generalize your units. Simplify your scale bar too. Same comments for other map scales.
Done, indeed very remarkable point, we checked the paper carefully and revised as you indicated.
Rename some of the units, garden could be recreational land? Maybe merge Pasture 1 and 2.
Thank you! The units checked again. However, most of the gardens are for apple, nuts, grape and are totally differ from pasture
Enlarge the labels on the map or make them stand out; currently they are difficult to read.
Done!
Figure 4: This figure is useless unless you provide the generalized name or provide the broader class for the units. Currently, the map has too many units and are too difficult to distinguish at this scale. So simply the map.
Indeed, very good points, this was difficult to us too, we did our best to simplify it, however the study area has very complex geology seething and in order to show the impacts of geology steeling on groundwater quality, honestly, we need such a figure. so wide range of variations are natural and simplifying the figure may affect the results. But we did our best to be obey for your comment and we hope that it is not offend your valuable comment.
Page 5: Last line: Correct the formatting (sub/superscript) for the ionic charges. Same comment for section 3.3 and all across the manuscript.
All of them checked. Thank you.
Section 3.2: What kinds of satellite data? Provide a few details.
Many thanks for checking the paper very carefully, we used the Landsat Satellite images and the image processing method was the integrated approach of Fuzzy Object Based Image analysis and Deep learning. Since we have taken this data from our early papers in this area, then we add a citation. But we also added further details in the new version of the paper.
Section 3.3: Correct the subscript formatting for the variables. Correct those all across the manuscript where they appear, this applies to other variables too.
Done!
Section 4.2: Second paragraph: First line-grammar?
Corrected.
Figure 6: Anion label needs to be fixed for all the graphs. You have not explained what is “SAR”. Also is it “TH” or “Th”?. Please maintain consistency with your symbols. Please expand your caption a little bit, it is confusing to look at first. What does the red cross-hair symbol represent? Explain.
All of them checked and revised. We also added this statement to figure caption to makes it easy the understanding of figure context Note: SAR stands for Sodium Adsorption Ratio, TH indicate the Total Hardness, TDS represent the total dissolved solids and (EC shows the conductivity
Figure 7: label Y-axis.
Revised to .tif format.
Table 3: What do those red values indicate? Maybe change the red color to italics or something since you cannot have a colored text on the table.
Yes. You are right. The red colors indicate to the correlation coefficient of r ≥ 0.75 which is changed to italic. Table 3 revised.
Figure 8 and 9: Legend not clear. What is “Aq cs”? Also, what does those numbers indicate? Explain in the caption.
The figure upgraded. The numbers are shown wells number (the wells named by number).
Figure 10: The graphs need to be cleaned a little bit, difficult to read. Maybe you can change the blue filled circles to empty ones to make it more readable?
Many thanks for checking our work very carefully, as mentioned above, we had large number of data, so we filter them before and this figure shows the cleaned well.
Page 20: Under Maragheh Aquifer:
Done!
Correct grammar for the first line.
Done
Last line of the paragraph, missing period
Done
Last paragraph, 5th line, typo: space needed.
Done
Correct the grammar for the line “SO42− and Cl− were like salinity in that they also have a positive correlation……” Done
In the same paragraph: last line starting with “When comparing the overall groundwater ….”
Done
It is confusing, which season has better quality, wet or dry? The sentence is convoluted, please rephrase it.
Indeed, revised essentially!
Finally, once more thanks you very much for the valuable comments. We did our best to apply your comments and we hope that you will find the revised version valuable to get published.
Authors

Reviewer 3 Report
The authors investigated the impact of climate in Urmia-Lake Basin using hydro-geochemical parameters of nearby aquifers. At first glance, the paper is way too long with a lack of critical discussion. The bulk of the content is overly-descriptive and repetitive, hence, the focus is not clear. The source data is from another source, so the data can be moved to the supplementary information. Below are my specific comments/suggestion:
- Abstract - "As groundwater is extracted..., contaminants become concentrated..." - this sentence is not correct. First, "contaminants" is not defined. Second, dissolved contaminants will be removed together with the extracted groundwater. Authors used "contaminants" loosely throughout the article, and this term should be defined properly.
- Introduction - 20 and 53 km3
- Introduction - "Excessive pumping can cause saltwater... to migrate upwards..." - Explain.
- Introduction - "Previously, several hydrology and hydro-geochemistry studies... [8, 13, ...]. However, no systematic investigation... along the eastern shore of Urmia Lake." - Why is there a need? Briefly discuss the relevant works cited.
- Introduction - "The Principal Component Analysis (PCA) has recently..." The authors listed 9 references, please choose 3-5 more significant ones. There is no need to cite all the works available.
- Introduction - "Groundwater quality deterioration is linked..." - same meaning as previous sentence.
- Study Area - "Urmia Lake Basin (ULB) is a center...." - same meaning as previous sentence. What is the focus of this work, the lake or basin?
- Study Area - "Climate change has led to drought..." - conflicts with rainfall data in Fig 2. Authors may want to use Fig 2 to explain this.
- 1 billion and 200 million = 1.2 billion
- The legend in Fig 4 is abbreviated, impossible to understand.
- Data Analysis - "Squared cosines of variable reflect the representation quality of a variable..." What does "representation quality" mean?
- Result - Tasuj, Tabriz, Tabriz - what are they?
- Groundwater hydrochemistry - hydrochemistry refers to the water chemistry, the authors only listed the water quality - "groundwater quality" is more appropriate. Should use a summary table instead of Fig 6. Rather than describing the trend, should discuss the significance of key parameters. Fig 6 should be moved to SI - the source is not mentioned.
- TDS and EC are dependent on each other.
- The numbers should have 0 dp, the decimal places has no meaning.
- Groundwater Quality Analysis - "good water quality" should be defined. What is cut-off value?
- Fig 8 and 9 should be Fig 8a) and b). The figure can be elaborated further. Discuss the differences between dry and wet seasons.
- Table 2 - 356 observations - for each aquifer or total?
- Table 3 - anion, TH, SAR - what are they exactly?
- Individual aquifers were discussed - most of the content was descriptive and repetitive. Suggest the authors to discuss all aquifers with special attention on specific aquifer if required. The general readers do not need to know each aquifer.
Author Response
Reviewer #3
Dear Reviewer,
We very much appreciate of your positive statements regarding our manuscript. It appears us you kindly spent significant time on this manuscript and we are grateful for the detailed comments. In fact, you brought up interesting aspects and we believe that these comments and our respective reactions to them will improve the quality of our paper. We did our best to improve the scientific quality of the manuscript significantly. Based on your constructive comments on early version and the comprehensive revisions on the manuscript, we are very confident that you will find this revised version now worthwhile to get published.
Comments to the Author
Dear reviewer, thank you for your comments in enhancing the scientific level of the paper. The main aim of the study is: 1- to delineate the aquifers condition in the eastern plains of Urmia Lake and the spatial distribution of potential information extracted from hydrogeochemical data by combining PCA with GIS. This can provide the better hydrogeochemical understanding of the multiple aquifers needed to take preventive action against groundwater damage. 2- to evaluate saltwater (Lake water) intrusion areas and high vulnerability zones of the aquifers 3- to highlight the suitability of groundwater for irrigation and drinking purposes for sustainable agriculture and basic human needs. Although there is a large data set, we had to examine all nearby and affected aquifers around the lake to meet our goals. Although the content are seems to be similar, each session is included critical information for natives. This paper is attempt to give a wide attitudes to the farmers and decision makers to make proper decisions.
Abstract - "As groundwater is extracted..., contaminants become concentrated..." - this sentence is not correct. First, "contaminants" is not defined. Second, dissolved contaminants will be removed together with the extracted groundwater. Authors used "contaminants" loosely throughout the article, and this term should be defined properly.
We revised it, it might also worth to mention that the term of contamination which is considered in this study is refer to the presence of a substance where it should not be or at concentrations above background but is not yet proven that it can result in adverse biological effects to resident communities.
Introduction - 20 and 53 km3.
Corrected, many thanks for checking the paper very carefully
Introduction - "Excessive pumping can cause saltwater... to migrate upwards..." - Explain.
Revised, we added further detail
Introduction - "Previously, several hydrology and hydro-geochemistry studies... [8, 13, ...]. However, no systematic investigation... along the eastern shore of Urmia Lake." - Why is there a need? Briefly discuss the relevant works cited.
As mentioned before, Urmia Lake is the largest saline lake in Iran which is located among the three province of East Azerbaijan, West Azerbaijan and Kurdestan. There are a few studies carried out in a very small area around the lake which cited in this paper. However, there is no such comprehensive study in Urmia lake basin with a broad view of the characteristics, qualitative changes and the linkage of the aquifers around the salt lake. Also, we could prepare groundwater quality maps during dry and rainy seasons for the first time in the study area. We added a further detail to cover your valuable comment.
Introduction - "The Principal Component Analysis (PCA) has recently..." The authors listed 9 references, please choose 3-5 more significant ones. There is no need to cite all the works available.
Done, we reduced the citation as you indicated
Introduction - "Groundwater quality deterioration is linked..." - same meaning as previous sentence.
Thank you for checking the paper very carefully, We removed it.
Study Area - "Urmia Lake Basin (ULB) is a center...." - same meaning as previous sentence. What is the focus of this work, the lake or basin?
This study was conducted in basin scale, Urmia Lake Basin (ULB) which is highly affected by drought of the Urmia Lake. however, we address basically the lakes drouth impacts on the salinization of nearby aquifers, so there are somehow related.
Study Area - "Climate change has led to drought..." - conflicts with rainfall data in Fig 2. Authors may want to use Fig 2 to explain this.
Indeed, Revised, actually the climate change together with intensive land use change by means of extension of the farmlands has resulted this critical situation of the lake, we added a further details to cover your valuable comments.
1 billion and 200 million = 1.2 billion.
Done.
The legend in Fig 4 is abbreviated, impossible to understand.
Revised, we have simplified the figure. However, we may indicate that they are geological abbreviations which are common in geology maps. They explain the geological characteristics of one unit. As the explanation of each of them is long, geologists have define specific abbreviations for them. For example: PlQc refers to Fluvial conglomerate, Piedmont conglomerate and sandstone for Pliocene-Quaternary. To be perfectly honest with you, the paper is too long now and revising it according to your comments and other 3 reviwers made it even much longer and we kindly ask you let us to have without much further details on geology since it is not the objective of this research. We just to indicated the situation of Geology setting in this area and its possible impacts on the groundwater salinization that we addressed in discussion section.
Data Analysis - "Squared cosines of variable reflect the representation quality of a variable..." What does "representation quality" mean?
Squared cosine indicates the quality of representation of the variable on the principal component. For example, a high Squared cosine indicates a good representation of the variable on the principal component. In this case the variable is positioned close to the circumference of the correlation circle. A low Squared cosine indicates that the variable is not perfectly represented by the PCs.
Result - Tasuj, Tabriz, Tabriz - what are they?
They are the local names of the aquifers which are taken from the Geographical locations.
Groundwater hydrochemistry - hydrochemistry refers to the water chemistry, the authors only listed the water quality - "groundwater quality" is more appropriate. Should use a summary table instead of Fig 6. Rather than describing the trend, should discuss the significance of key parameters. Fig 6 should be moved to SI - the source is not mentioned.
Many thanks for the valuable comment, honestly Fig. 6 is very critical figure in the manuscript. the scatter plots of the data which included useful information as mean, Max, Min, mode, range of each parameter and make it easy to see all the information together. As the data set was large in this study, we came to the conclusion that the scatter plot is the proper way to show the maximum information together. All the values are in SI system in this manuscript and their units have been mentioned in the text.
TDS and EC are dependent on each other.
Indeed, thanks for indicating it!
The numbers should have 0 dp, the decimal places has no meaning.
We checked the paper and removed unnecessary extra values.
Groundwater Quality Analysis - "good water quality" should be defined. What is cut-off value?
Thanks, We determined the classes of water quality based on the results of WQI. We also added the following table as you indicted.
WQI |
Type of water |
Less than 50 |
excellent |
50-100 |
good |
100-200 |
poor |
200-300 |
Very poor |
More than 300 |
Unsuitable for drinking purposes |
Fig 8 and 9 should be Fig 8a) and b). The figure can be elaborated further. Discuss the differences between dry and wet seasons.
Yes, it is true, we added further details. however, since we also wanted to observe the impacts of season on QWI, so we decided to have them separately. We hope it is fine with you. Honestly, we also used them in our discussion and elaborating them dose not that much supports our justification in the paper. We really hope it is alright with you.
Table 2 - 356 observations - for each aquifer or total?
It was for total observation.
Table 3 - anion, TH, SAR - what are they exactly?
Revised, we also added this to the table, Sodium Adsorption Ratio =(SAR), TH= Total Hardness and added to the table
Individual aquifers were discussed - most of the content was descriptive and repetitive. Suggest the authors to discuss all aquifers with special attention on specific aquifer if required. The general readers do not need to know each aquifer.
Many thanks for the valuable comment, we did our best to addressee it, we may also indicate that honestly this paper is attempt to give a wide attitudes to the farmers and decision makers to make proper decisions. Although the content are seems to be similar, each session is included critical information for natives.
Finally, once more thanks you very much for the valuable comments. We revised a paper according to your comments and the other 3 reviewers, honestly, we did our best we hope that you will find the revised version valuable to get published.
Authors

Reviewer 4 Report
1 PLS clearly state the research purposes for their manuscript.
2 More further explains for their weight assignments in Table 1 is essential。
3 My personal point of view, I could not see “a decreasing trend was observed from the Shiramin aquifer to the south of the basin” mentioned under Section 4.1 Groundwater Hydrochemistry.
4 The authors emphasized the seasonal variation of water quality parameters for aquifer under Section 4.2 Groundwater Quality Analysis. Why is the seasonal variation ignored in the Spearman correlation analysis and PCA analysis?
5 There are too many problems in their figure s and tables. PLS double check
Author Response
Dear Reviewer,
We very much appreciate of your positive statements regarding our manuscript. It appears us you kindly spent significant time on this manuscript and we are grateful for the detailed comments. In fact, you brought up interesting aspects and we believe that these comments and our respective reactions to them will improve the quality of our paper. We did our best to improve the scientific quality of the manuscript significantly. Based on your constructive comments on early version and the comprehensive revisions on the manuscript, we are very confident that you will find this revised version now worthwhile to get published.
Comments and reply
1 PLS clearly state the research purposes for their manuscript.
Revised, we added the research objective and its contribution to the end of introduction section, we have several objectives by this study, in brief the main aim of the study is: a) to delineate the aquifers condition in the eastern plains of Urmia Lake and the spatial distribution of potential information extracted from hydrogeochemical data by combining PCA with GIS. This can provide the better hydrogeochemical understanding of the multiple aquifers needed to take preventive action against groundwater damage. B) to evaluate saltwater (Lake water) intrusion areas and high vulnerability zones of the aquifers 3- to highlight the suitability of groundwater for irrigation and drinking purposes for sustainable agriculture and basic human needs.
2 More further explains for their weight assignments in Table 1 is essential。
Thanks for the comment, revised and more details and a reference for more information are added in the text.
3 My personal point of view, I could not see “a decreasing trend was observed from the Shiramin aquifer to the south of the basin” mentioned under Section 4.1 Groundwater Hydrochemistry.
The trend analysis is not conducted in this study and we concluded this result by evaluating the mean values. Anyhow we agree that it is not an obvious result. That is why the mentioned sentence was removed from the text.
4 The authors emphasized the seasonal variation of water quality parameters for aquifer under Section 4.2 Groundwater Quality Analysis. Why is the seasonal variation ignored in the Spearman correlation analysis and PCA analysis
Thanks a lot for very detailed comment. Seasonal variation and correlation analysis were conducted in this area and their results are summarized in Fig. 10. For example 322 (w) in Fig.10 stands for the well with the code of 322 in the wet season. They all are presented in detail in the result section.
e.g. A closer look at Fig. 10 reveals that the factors with scores of less than F1<-2.5 (wells 6 (dry season), 8 (dry season), 10, 48, 60 (dry season), 93, 122, 123, 124, 125, 126, 127, 128, 167, 181, 195 (dry season), 272 (dry season), 324 (dry season), 325 and 326) have fresh water with low mineralization. In contrast, the groundwater of the wells (26, 113, 154, 162, 177, 185, 213 (wet season), 214, 242 (wet season), 257, 264, 288 (wet season), 320, 403, 429, 430 (wet season) and 435 (wet season)) with factor scores of more than F1>4.5 are highly mineralized, with TDS values between 3172.4875 and 7525.375 mg/L and total hardness ranging between 1444.33 and 1271 mg/L.
5 There are too many problems in their figure s and tables. PLS double check.
Revised, we have checked all manuscript carefully and as you as we have revised it substantially.
Finally, once more thanks you very much for the valuable comments. We revised a paper according to your comments and the other 3 reviewers, honestly, we did our best we hope that you will find the revised version valuable to get published.
Authors

Round 2
Reviewer 3 Report
The authors have made some amendments to the review comments; however, the main problem with the manuscript is the length of the paper, and it has not been addressed. The discussion remained overly descriptive, and the authors could not highlight major differences among the aquifers (pg 19-23). Fig. 6 is extremely difficult to read and can be summarized into a table instead, and some of the scales are obviously wrong. Fig. 10 seemed to contain a lot of content, but the discussion was so brief that the figure did not seem important. Authors should seriously consider the importance and prioritize the content. The authors also used numerous analytical tools to analyze the data, but the rationale for doing so was not explained. Rather than treating each analysis as an independent event, authors should try to relate the results of each analysis. I also do not understand why we need to use so many analytical tools to analyze the data.
Author Response
Reviewer #3
Dear Reviewer,
We very much appreciate of your valuable time for double checking this manuscript. We revised a paper once again according to your comments and we hope that you will find the new revised version satiable to be accepted and published.
Comments to the Author
The authors have made some amendments to the review comments; however, the main problem with the manuscript is the length of the paper, and it has not been addressed. The discussion remained overly descriptive, and the authors could not highlight major differences among the aquifers (pg 19-23).
Many thanks for the valuable comments, we have checked the manuscript one again carefully based on your valuable comments. As you see from the revised version, we removed the unnasty section are removed and manuscript shortened accordingly. The manuscript size from 11250 has reduced to 10200 and we removed about 1000 words. Honestly since we apply and compare of the aquifers quality, then the size of manuscript is ultimately being increased. As you kindly recommended we reduced it accordingly and we really hope it is fine with you now.
Fig. 6 is extremely difficult to read and can be summarized into a table instead and some of the scales are obviously wrong.
Fig. 6 includes useful information as mean, Max, Min, mode, ranges of 15 water quality parameters in two seasons in the 8 aquifers. As all these information could not gather into one table, we chose the scatter plot to present all the information together. As the data set was large in this study, we came to the conclusion that the scatter plot is the proper way to show the maximum information together.
Fig. 10 seemed to contain a lot of content, but the discussion was so brief that the figure did not seem important.
Fig. 10 includes the results of PCA, CA and WQI clustering in one shot. All the results presented in the pages of 18-22 for each studied aquifer separately are extracted from this figure.
Authors should seriously consider the importance and prioritize the content. The authors also used numerous analytical tools to analyze the data, but the rationale for doing so was not explained. Rather than treating each analysis as an independent event, authors should try to relate the results of each analysis. I also do not understand why we need to use so many analytical tools to analyze the data.
Actually, all analytical tools we have used are: 1) PCA to summarize the large data set and extract more important parameters which have more significant roles in water quality in the study area. PCA simplifies the complexity in high-dimensional data. It does this by transforming the data into fewer dimensions. PCA helps us interpret our data. 2) We calculated the WQI to determine the water status in terms of the quality. 3) CA to group similar parts in the aquifer in terms of water quality (based on the results of WQI) and to cluster the data into structures that are more easily understood. 4) GIS to visualize our results. GIS helps users understand patterns, relationships, and geographic context.
Finally, once more thanks you very much for the valuable comments. We revised a paper according to your comments, honestly, we did our best we hope that you will find the revised version valuable to get published.
Authors

Reviewer 4 Report
1 PLS explain further how the weights assigned.
2 The research period is 10 years, and the land use used by the author is 2020. Please further explain why select the land use in 2020 ?
3 Did the authors consider the changes in water quality data over a 10-year period when calculating the WQI value?
Author Response
Reviewer
Dear Reviewer,
We very much appreciate of your valuable time for double checking this manuscript. We revised a paper once again according to your comments and we hope that you will find the new revised version satiable to be accepted and published.
1 PLS explain further how the weights assigned.
The general structure of WQI is:
1) Selection of the water quality parameters: one or more water quality parameters are selected for inclusion in the assessment
2) Generation of the parameter sub-indices: parameter concentrations are converted to unit less sub-indices
3) Assignment of the parameter weight values: parameters are assigned weightings depending on their significance to the assessment
4) Computation of the water quality index using an aggregation function: the individual parameter sub-indices are combined using the weightings to give a single overall index.
Two approaches have been commonly used for obtaining appropriate parameter weight values. Many WQI models used expert opinion to inform the parameter weighting process. And some of them used Multi decision making tools like as Analytic Hierarchy Process (AHP).
We used from regional guideline (IRWQIG) for employing the WQI model in this study. This guideline uses unequal parameter weight values which sum to 1. The original weight values were obtained by employing an expert panel. Environment significance and relative impacts were considered for giving weight values. Expert panels typically comprise key stakeholders such as water quality experts, policymakers or practitioners, government representatives and nongovernmental organizations or authorities responsible for managing water resources quality. But subsequent applications of the model have used modified weight values (employing decision making tools of Analytic Hierarchy Process (AHP)) for evaluating surface water quality. The AHP technique has been used to determine parameters significance and therefore reduces uncertainty resulting from inappropriate weighting of parameters. The analytic hierarchy process (AHP) method was developed by Thomas Saaty in the 1970 s. It is a technique for decision making in complex environments in which many variables or criteria are considered in the prioritization and selection of alternatives. In the context of WQI parameter weightings, it allows one to determine the most appropriate weightings for given parameters that are reflective of their influence on overall water quality. This helps to check the reliability of the decision maker’s assessments, and it also reduces preconceptions in the decision-making process. Several scientists have noted that AHP is an effective method that can minimize model uncertainty and increase the accuracy of the weighting procedure (Sarkar and Abbasi, 2006; Ocampo-Duque et al., 2006; Gazzaz et al., 2012 and Sutadian et al., 2017)
Then we employed cluster analysis to classify the water quality based on the overall index value.
- Sarkar, C., Abbasi, S.A., 2006. Qualidex - A new software for generating water quality indice. Environ. Monit. Assess. 119, 201–231. https://doi.org/10.1007/s10661-005- 9023-6.
- Ocampo-Duque, W., Ferr´e-Huguet, N., Domingo, J.L., Schuhmacher, M., 2006. Assessing water quality in rivers with fuzzy inference systems: A case study. Environ. Int. https://doi.org/10.1016/j.envint.2006.03.009.
- Gazzaz, N.M.N.M., Yusoff, M.K.M.K., Aris, A.Z.A.Z., Juahir, H., Ramli, M.F.M.F., 2012. Artificial neural network modeling of the water quality index for Kinta River (Malaysia) using water quality variables as predictors. Mar. Pollut. Bull. 64, 2409–2420. https://doi.org/10.1016/j.marpolbul.2012.08.005.
- Sutadian, A.D., Muttil, N., Yilmaz, A.G., Perera, B.J.C., 2017. Using the Analytic Hierarchy Process to identify parameter weights for developing a water quality index. Ecol. Indic. 75, 220–233. https://doi.org/10.1016/j.ecolind.2016.12.043. Sutadian, A.D., Muttil, N., Yilmaz, A.G., Perera, B.J.C., 2018. Development of a water
- Tomas, D., Curlin, ˇ M., Mari´c, A.S., 2017. Assessing the surface water status in Pannonian ecoregion by the water quality index model. Ecol. Indic. 79, 182–190. https://doi. org/10.1016/j.ecolind.2017.04.033.
The research period is 10 years, and the land use used by the author is 2020. Please further explain why select the land use in 2020?
Many thanks for the comments, actually we used the Land use 2020 as we wanted to have most recent information regarding the current land use/cover of the study area. In addition, we have time series land use/cover mapping of the study area from 1990 to 2020 and as we also referenced in the manuscript to our early papers regarding the land use/cover of this area. As we had access to accurate land use/cover map of the study area, then we decide to use it as most current data in time of research implementation we really hope it is fine with you.
Did the authors consider the changes in water quality data over a 10-year period when calculating the WQI value?
That is right, thanks for the comment.
Finally, once more thanks you very much for the valuable comments. We revised a paper according to your comments and the other 3 reviewers, honestly, we did our best we hope that you will find the revised version valuable to get published.
Authors
